# Impact of aerosols and turbulence on cloud droplet growth: An in-cloud seeding case study using a parcel-DNS approach

Sisi Chen[12], Lulin Xue[13], and Man-Kong Yau[2]

[1]National Center for Atmospheric Research, Boulder, Colorado, USA
[2]McGill University, Montréal, Québec, Canada
[3]Hua Xin Chuang Zhi Sci. & Tech. LLC, Beijing, China

**Correspondence:** Sisi Chen (sisichen@ucar.edu)

**Abstract.**

This paper investigates the relative importance of turbulence and aerosol effects on the broadening of the droplet size distribution (DSD) during the early stage of cloud and raindrop formation. A parcel-direct numerical simulation (DNS) hybrid approach is developed to seamlessly simulate the evolution of cloud droplets in an ascending cloud parcel. The results show that turbulence and CCN hygroscopicity are key to the efficient formation of large droplets. The ultra giant aerosols can quickly form embryonic drizzle drops and thus determines the onset time of autoconversion. However, due to their scarcity in natural clouds, their contribution to the total mass of drizzle drops is insignificant. In the meantime, turbulence sustains the formation of large droplets by effectively accelerating the collisions of small droplets. The DSD broadening through turbulent collisions is significant and therefore yields a higher autoconversion rate compared to that in a non-turbulent case. It is argued that the level of autoconversion is heavily determined by turbulence intensity. This paper also presents an in-cloud seeding scenario designed to scrutinize the effect of aerosols in terms of number concentration and size. It is found that seeding more aerosols leads to higher competition for water vapor and reduces the mean droplet radius, and therefore slows down the autoconversion rate. On the other hand, increasing the seeding particle size can buffer such negative feedback. Despite that the autoconversion rate is prominently altered by turbulence and seeding, bulk variables such as liquid water content (LWC) stays nearly identical among all cases. Additionally, the lowest autoconversion rate is not co-located with the smallest mean droplet radius. The finding indicates that the traditional Kessler-type or Sundqvist-type autoconversion parameterizations which depend on the LWC or mean radius could not well-capture the drizzle formation process. Properties related to the width or the shape of the DSD are also needed, suggesting that the Berry-and-Reinhardt scheme is conceptually better. It is also suggested that a turbulence-dependent relative-dispersion parameter should be considered.

# 1 Introduction

Aerosol-cloud-precipitation interactions represent one of the major uncertainties in weather and climate prediction (Fan et al., 2016). Current atmospheric models can not resolve the microphysical processes and thus rely on parameterizations to represent those interactions. Studies show that model results of the location and intensity of precipitation are sensitive to microphysics schemes (Xue et al., 2017; White et al., 2017; Grabowski et al., 2019). For example, White et al. (2017) showed that the autoconversion scheme is the dominant factor to account for the difference in rain production, and the uncertainty due to the choice of microphysical parameterizations exceeds the effects of aerosols. No benchmark "truth" from either measurements or modeling exists to gauge the performance of various microphysics schemes. On the one hand, in-situ measurements cannot directly obtain the process rates, such as the rate of autoconversion and accretion, which prevents such microphysical processes from being accurately modeled (Morrison et al., 2020). The community has to rely on laboratory experiments, indirect observations, or theoretical models to develop and validate microphysical schemes (e.g., Stoelinga et al., 2003; Wood et al., 2002; Wang et al., 2005). On the other hand, laboratory facilities such as cloud chambers are difficult to create environments scalable to real clouds. Furthermore, the effects of chamber walls, such as the heat and moisture fluxes fed into the solid wall and the droplet loss due to their contact with the wall, are challenging to quantify with considerable uncertainties in the measurements (e.g. Thomas et al., 2019).

In this study, we implement the idea of in-cloud seeding, i.e., seeding hygroscopic particles near the cloud base to investigate the effects of aerosols in droplet growth and rain formation. Hygroscopic cloud seeding, owing to its potential effect of increasing rainfall, has been conducted in research and operational context globally to address the shortage of water resources in arid environments (e.g., Silverman and Sukarnjanaset, 2000; Terblanche et al., 2000). The general concept of hygroscopic cloud seeding in rain enhancement is that the introduction of artificial cloud condensation nuclei (seeding particles) into warm clouds can, on the one hand, suppress the activation of small natural aerosols, and on the other hand, generate large initial particles that accelerate or enhance the collision-coalescence process (Cooper et al., 1997). Regardless of its existence in operational weather modification for decades, the direct effect of seeding is still inconclusive, partly due to the chaotic nature of the convective cloud system making it impossible to conduct controllable seeding experiments and the limitation in detecting and assessing the seeding results with current instrumentations (Silverman, 2003; Flossmann et al., 2019). Nevertheless, the progress made in cloud seeding does advance our understanding of cloud-aerosol-precipitation interactions. A leading idea of this study is to make use of the concept of cloud seeding experiments to separate the influence of aerosols on rain initiation from the effects of natural cloud processes such as turbulence and aerosol hygroscopicity, as well as to shed light on the long-existing question of whether cloud seeding could enhance precipitation.

Currently, direct numerical simulation (DNS) is believed to be the only numerical approach capable of simulating the growth of individual cloud particles in turbulent flows (Grabowski et al., 2019). Only a few DNS studies to date investigated the evolution of the droplet size distribution (DSD) in an updraft environment (e.g., Chen et al., 2018b; Gotoh et al., 2016; Saito and Gotoh, 2018). However, the solute effect (aerosol hygroscopicity) and curvature effect were excluded in those works for simplicity. Parcel model studies on droplet condensation in a lifted parcel show that the curvature term and the solute term

can lead to condensational broadening on the droplet size spectrum. Srivastava (1991) demonstrated that the curvature effect is essential for DSD broadening in an ascending parcel. Korolev (1995) found that the curvature effect and the solute effect lead to irreversible broadening when supersaturation fluctuations are present. It is also found that aerosols of different sizes and different hygroscopicity can cause spectral broadening without supersaturation fluctuations (Çelik and Marwitz, 1999; Jensen and Nugent, 2017). Therefore, it is crucial to examine whether these effects are important in spectral broadening when they dynamically couple with droplet collisional growth in a turbulent environment.

It is recognized that DNS is computationally expensive. To achieve an accurate representation of cloud microphysics while maintaining a feasible computational load, a hybrid modeling framework that combines a parcel model and a DNS model is proposed in this study. The parcel model provides the mean state of the air parcel and can be used when the effect of turbulence is less prominent. The DNS model explicitly resolves all small-scale turbulent eddies which are key to cloud particle interactions. The Lagrangian particle-by-particle method is employed in the DNS to track the evolution of individual cloud particles coupling with the turbulent flow. This hybrid parcel-DNS approach allows a close examination of the growth history of cloud particles from aerosol activation to drizzle formation. By comparing simulations with different aerosol and turbulent conditions, we are able to evaluate the contribution of each microphysical component to warm rain initiation. The ultimate goal is to provide a numerical benchmarking tool to better understand aerosol-cloud-precipitation interaction at fine scales and improve the sub-grid-scale representation of clouds and precipitation in numerical weather and climate prediction.

Chen et al. (2018b) found that the evolution of DSD in turbulence is different depending on whether droplets grow by condensation-only, collision-only, or condensation-collision (Fig. 1 in their paper). This reveals that droplet condensation and collisions when interacting with turbulence, cannot be treated as the linear addition of the two processes. Many past DNS studies focused on either the condensation-only process or the collision-only process which might yield biased results. It should be pointed out that autoconversion defined as the mass transfer from small droplets to embryonic drizzle drops via collision-coalescence should not exclude the impact of condensational growth, as the two processes dynamically interact with each other.

This paper presents a sequel to the study of Chen et al. (2018b) by addressing several caveats mentioned in their paper. Firstly, Chen et al. (2018b) treated only pure water droplets as is commonly assumed in most DNS studies (e.g., Sardina et al., 2015; Vaillancourt et al., 2001, 2002; Paoli and Shariff, 2009). This simplification may underestimate the rate of droplet growth by condensation. Jensen and Nugent (2017) found that cloud condensation nuclei (CCN) strongly enhances the particle growth, and droplets with giant CCN can even grow in regions of sub-saturated downdrafts. In our new hybrid approach, we use an accurate droplet diffusional growth equation including both curvature effect and solute effect. Secondly, the initial DSD in Chen et al. (2018b) obtained from flight observations was a result of averages over a long-time period and along a long sampling path (including both core regions and cloud edges). The average might mask the local property of an adiabatic core that the DNS aims to simulate. The adiabatic cores are regions free of entrainment of dry air. This region has a higher liquid water content (LWC) than the rest of the cloud and is argued to favor the formation of raindrops (Khain et al., 2013). To represent the DSD evolution at the core region, we prescribe here a dry aerosol size distribution in the sub-cloud region, and

the aerosol activation process is explicitly simulated by a parcel model to provide a more physically-based initial DSD for the DNS.

The main purpose of the present study is to investigate the relative importance of turbulence, CCN hygroscopicity and aerosols (size and number concentration) on the DSD broadening in cumulus clouds. The paper is organized as follows. Sections 2.1-2.2 introduce the hybrid model of a parcel-DNS framework. In Section 2.3, the configuration of the 12 numerical simulations are described to compare the microphysical responses to turbulence (turbulent vs non-turbulent), hygroscopicity (pure-water droplets vs solute-containing droplets), aerosol size and number concentration (with or without seeding particles), and droplet growth mechanisms (condensation-only vs condensation-collision). Results are presented in Section 3, showing that turbulence and CCN hygroscopicity are key to the formation of big droplets, and seeding slows down the broadening and lowers the autoconversion rate. Summary and outlook for future work are in Section 4.

## 2    Model setup

A hybrid model is used in this paper for simulating the droplet growth inside an ascending cloud parcel. The ascent is divided into two phases based on the distinct dominant microphysical processes. A parcel model and a DNS model are combined to seamlessly simulate the two phases, as illustrated in the schematic diagram in Fig. 1. The first phase starts from the unsaturated sub-cloud region ($\approx 300\ m$ below cloud base) to the level where the supersaturation reaches a maximum ($\approx 43\ m$ above cloud base, see Fig. 2(a)). During this phase, supersaturation increases with height, and the microphysical process is dominated by aerosol activation. Cloud particles remain small and collisional growth is negligible. A non-turbulent parcel model is employed to calculate the droplet growth by condensation in this phase. The second phase starts from the level of maximum supersaturation (=1.59%) to $1\ km$ above which takes 500 s in simulated time (Table 1). At this stage, no new activation occurs as the supersaturation starts to decrease with height. This phase is dominated by cloud droplet growth. DNS model is employed to calculate individual droplet growth by condensation and collision affected by its immediate local turbulent environment. Parcel model state at the height with maximum supersaturation is fed into DNS as initial conditions. Because unactivated aerosols have little influence on the subsequent droplet growth or on the water vapor field, only the activated aerosols from the parcel model are carried over to the DNS model as the initial background aerosol condition to decrease the computational load. The CCN size distribution and droplet size distribution are displayed in Fig. 2(c). This parcel-DNS hybrid model provides an economical approach and is the first step towards a fully DNS-resolved simulation of the entire ascending process.

### 2.1    Parcel model

The parcel model is adopted from Jensen and Nugent (2017) with two main modifications: (1) The droplet collision-coalescence is excluded for simplicity because most particles in this phase are smaller than $10\ \mu m$. These droplets have very small collision rates even in strong turbulence (Chen et al., 2016, 2018a), and the growth is dominated by condensation. (2) The hygroscopicity

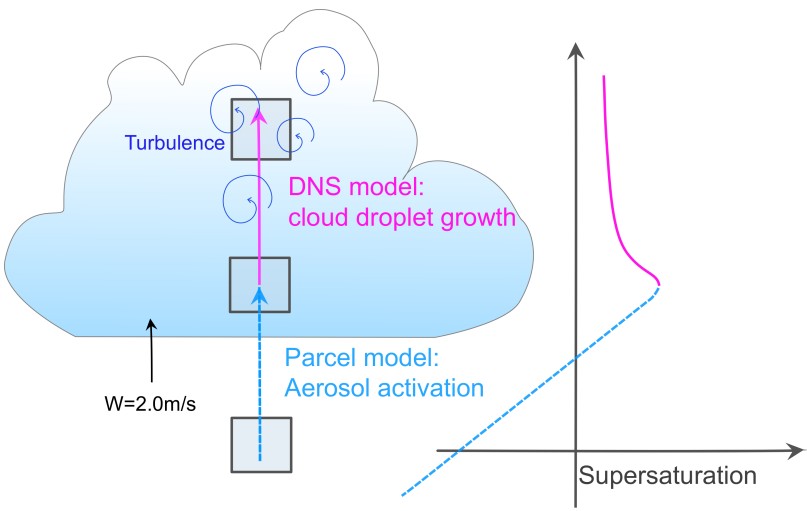

**Figure 1.** Schematic diagram of the parcel-DNS hybrid model along with the unscaled bulk supersaturation with height. The parcel model simulates the ascending process below the height of maximum supersaturation (dashed blue line), and the DNS simulates the subsequent ascending process (solid violet line).

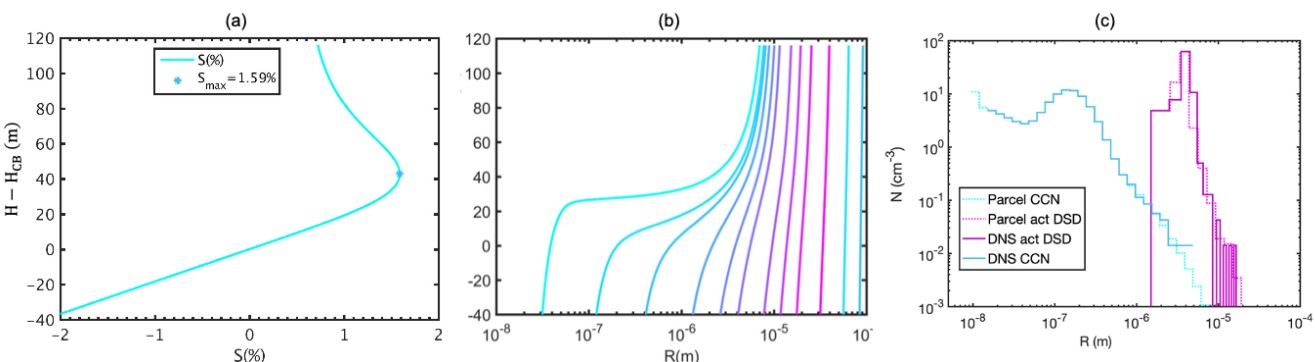

**Figure 2.** (a) Supersaturation and (b) radius of droplets with different initial wet sizes varying with the height from cloud base ($H - H_{CB}$). Only bins of activated particles are illustrated in (b). (c) The background natural CCN (dry particle) size distribution in the parcel model (light dotted blue histogram) and in the DNS model (darker solid blue histogram), and the droplet size distribution at maximum supersaturation ($S_{max} = 1.59\%$) in the parcel model (light dotted violet histogram) and in the DNS model (darker solid violet histogram). The vertical axis denotes the number concentration of the assigned particle size in the model.

parameter, $\kappa$, proposed by Petters and Kreidenweis (2007, their equation (6)) is employed in the droplet diffusional growth equation:

$$R\frac{dR}{dt} = \frac{S - \frac{R^3 - R_d^3}{R^3 - R_d^3(1-\kappa)} exp(\frac{2\sigma_w}{R_v \rho_w T R})}{\frac{\rho_w R_v T}{e_s D'} + \frac{\rho_w L_v}{K'T}(\frac{l_v}{R_v T} - 1)} f_v,$$
(1)

here $R$ is droplet radius, $R_d$ is the radius of CCN, $\sigma_w = 7.2 \times 10^{-2}\ Jm^{-2}$ is surface tension of water against air, $R_v = 467\ Jkg^{-1}K$ is individual gas constant for water vapor, $\rho_w$ and $\rho_a$ are the density of water and air, respectively, $T$ is air temperature, and $e_s$ is the saturated water vapor pressure. $D'$ and $K'$ are respectively the water vapor diffusivity and thermal conductivity that include kinetic effects (see equation (11a)-(11b) in Grabowski et al., 2011), and $L_v = 2.477 \times 10^6\ Jkg^{-1}$ is the latent heat of vaporization. $S$ is supersaturation ratio defined as $\frac{q_v}{q_{vs}} - 1$ where $q_v$ and $q_{vs}$ are water vapor mixing ratios at the current condition and at saturated condition, respectively. $f_v$ is ventilation coefficient which takes into account the distortion in water vapor field around the droplet surface when the droplet moves relative to the flow. Studies show that the effect is negligible when droplets are smaller than $10\ \mu m$ in radius (Rogers and Yau, 1989, p116). Therefore, the ventilation effect is excluded in this phase, i.e., $f_v = 1$. In DNS, we apply the empirical formulas of $f_v$ from Beard and Pruppacher (1971) which depends on the droplet Reynolds number and Schmidt number (see also equation (B2)-(B3) in Chen et al., 2018b).

There are two advantages of using the hygroscopicity parameter: 1) The chemical information of the aerosol (i.e., molecular weight, van Hoff factor, density, etc.) is simplified into a single parameter in the solute term; 2) the hygroscopicity parameter of mixed solute due to collision-coalescence can be simply calculated by a weighted average of the volume fractions of each component in the mixture (Petters and Kreidenweis, 2007).

The initial environmental conditions are taken from the cumulus cloud case of Jensen and Nugent (2017, Table 2). The parcel ascends from $H = 600\ m$ ($\approx 284\ m$ below cloud base) with a constant updraft velocity of $2.0\ ms^{-1}$, resembling a fair-weather cumulus cloud condition. The detailed information is listed in Table 1. The CCN (dry aerosol) size distribution fits a lognormal distribution, taken from the pristine case of Xue et al. (2010) (light blue histogram in Fig. 2(c)). The distribution consists of three log-normal modes in which the geometric mean dry radii in the three modes are $R = [0.0039, 0.133, 0.29]\ \mu m$, the geometric standard deviations are $\sigma = [4.5394 1.6218 2.4889]$, and the total number concentrations of the whole size range are $N = [133, 66.6, 3.06]\ cm^{-3}$. The initial size is discretized into 39 bins on a log scale with the bin width set by doubling the mass, or with a multiplication factor in radius of 2.0. In this way, the resolution is higher at small particle sizes and lower at large particle sizes. The bin size ranges from $0.006\ \mu m$ and $49\ \mu m$, which gives a total number concentration $N = 112\ cm^{-3}$. To examine the variation in the activation fraction of the aerosols due to bin width resolution, we performed sensitivity test with the number of bins spanning from 32 to 253, corresponding to a multiplication factor from 2.2 to 1.1. The result shows that the variation caused by changing the bin resolution has a decreasing trend with increasing resolution, with a maximum variation of 2.3% of the total aerosol number concentration in the 32-bin case. In particular, the 39-bin case has only $0.6\ cm^{-3}$ more aerosols activated than in the 253-bin case.

It is worth noting that the number concentration of CCN larger than $10\ \mu m$ is below $10^{-4}\ cm^{-3}$, corresponding to less than one particle in the DNS domain ($L = 16.5\ cm$). The hygroscopicity parameter of all aerosols is assumed $\kappa = 0.47$. The moving-bin method or moving-size-grid method (see discussion in Yang et al., 2018) is applied to calculate the evolution of the DSD. For aerosols with dry radius $R_d <= 1\ \mu m$, the initial wet radius is set to the size when the droplet is in equilibrium at the given ambient humidity: $dR/dt = 0$ (Jensen and Nugent, 2017). For giant aerosols with $R_d > 1\ \mu m$, the initial wet size is assumed to be twice the dry volume, i.e., $R = 2^{1/3}R_d$. As illustrated in Fig. 2(b), the droplets with initial radius below $1\ \mu m$

grow quickly by condensation between $20 - 40\ m$ above the cloud base before the maximum supersaturation is reached, and droplets larger than $1\ \mu m$ grow slower, creating a narrow DSD near the cloud base.

**Table 1.** Model description and initial conditions of the parcel model and the DNS model.

|  | Parcel | DNS |
|---|---|---|
| | Model description | |
| Domain size | 0D air parcel | $0.165 \times 0.165 \times 0.165\ m^3$ |
| $\Delta x$ | - | $1.289 \times 10^{-3}\ m$ |
| $\Delta t$ | $10^{-4}\ s$ | $3.15 \times 10^{-5}\ s$ |
| Microphysics treatment | Moving-bin method | Lagrangian particle-by-particle method |
| | Initial conditions | |
| Initial temperature | 284.3 K | 281.2 K |
| Initial pressure | 938.5 hPa | 902.2 hPa |
| Initial number concentration of natural background aerosols | $112\ cm^{-3}$ | $85\ cm^{-3}$ |
| Initial saturation ratio | 85.61% | 101.59% |
| Updraft velocity | $2.0\ m\ s^{-1}$ | $2.0\ m\ s^{-1}$ |
| Simulated time | 300 s | 500 s |

## 2.2 DNS model

All DNS simulations are initialized with an identical mean state listed in Table 1. A constant mean updraft speed of $2\ m\ s^{-1}$ is prescribed to lift the air parcel. The initial mean-state variables for DNS are obtained from the parcel model output at maximum supersaturation ($S = 1.59\%$). Above this altitude, no further activation is expected in the parcel due to the decreasing supersaturation. The inactivated aerosols, corresponding to the first two bins of the light blue histogram in Fig. 2(c), do not influence the subsequent evolution of the DSD. Therefore, only the activated aerosols from the parcel model are carried over

to the DNS, reducing the particle number concentration to $N = 85\ cm^{-3}$. This treatment avoids the computation of tracking the inactivated particles. In the parcel model, the droplet size is calculated by using the moving-bin method. The dry radius of each bin remains constant, and the wet radius grows by condensation. To assign the initial droplet size and its dry radius in the DNS, we regrouped the activated droplet bins into 15 droplet size groups ($R = 2 - 16\ \mu m$) with an interval of $1\ \mu m$. Their CCN sizes remain the original value. Due to the parallelization setup in the model, the initial number of each droplet size group

has to be an exact multiple of the number of processors in the simulation (64 processors are used in the present simulations). Therefore, a small difference in the resulting DSD between the two models is expected, as shown in Fig. 2(c).

The DNS model in the present study is initially developed by Vaillancourt et al. (2001) and has undergone a few modifications since then (Franklin et al., 2005; Chen et al., 2016, 2018a, b). The model employs two sets of equations: 1) the macroscopic equations to calculate the base-state (bulk) variables, and 2) the microscopic equations to calculate the fluctuation of the

175 variables affected by the small-scale turbulence and the local droplet condensation. A detailed description of the DNS model can be found in Chen et al. (2018b, Section 2 and Appendix B).

Two modifications are made in the present study. First, we use equation (1) to replace the simplified version of the droplet growth equation in Chen et al. (2018b, equation (B1)) where the curvature term and the solute term are excluded. Second, droplets with $R < 5\ \mu m$ are treated as non-inertial particles due to their small Stokes number, i.e., their velocity is equal to

180 the flow velocity. The length of a timestep is constrained by the inertial response time of the smallest inertial particle (see discussion in Chen et al., 2018a, on the length of the timestep). The treatment above avoids using too small a timestep when small droplets are present. For droplets between $5 - 40\ \mu m$, their motion is determined by both the Stokes drag force and gravity, and for droplets over $40\ \mu m$ nonlinear drag force is considered (see full description below the equation (B10) in Chen et al., 2018b). Droplets over $50\ \mu m$ are treated as fall-out and are removed from the simulation.

## 2.3 DNS experimental design

Two sets of experiments are performed. Each set consists of six cases, which gives 12 simulations in total. The first set of the experiments includes both condensational and collisional growth of droplets and will be referred to as the "condensation-collision" set. The second set excludes the droplet collision and will be referred to as the "condensation-only" set. The model setup for the two sets is the same other than the difference mentioned above. The configuration of the six cases is listed in Table

2. We focus on the condensation-collision set in the result section unless explicitly specified, and the condensation-only set is for the purpose of comparison to evaluate the influence by condensation and collision-coalescence.

**Table 2.** Model configuration of the six cases in each set of the experiment. Two sets of experiments are performed: set one includes both collision and condensation in the droplet growth and is referred to as the "condensation-collision" set; set two only considers droplet condensation and is referred to as the "condensation-only" set. This gives 12 cases in total. The natural DSD is taken from the parcel model output at $S = 1.59\%$. Monodisperse seeding is considered in "seeded" cases with CCN size ($R_d$) and initial droplet size ($R$) listed in the table.

|                | Experiments    | Turbulence | Solute effect | Initial DSD                                                                      |
| -------------- | -------------- | ---------- | ------------- | ------------------------------------------------------------------------------- |
|                | Run CTL        | on         | on            | Natural DSD                                                                      |
| Natural cases  | Run NoTurb     | off        | on            | Natural DSD                                                                      |
|                | Run NoSolu     | on         | off           | Natural DSD                                                                      |
|                | Run Seed-1N1R  | on         | on            | Natural DSD+ "seeding" particle ($R_d = 0.1\ \mu m$, $R = 4\ \mu m$, $N = 10\ cm^{-3}$) |
| "Seeded" cases | Run Seed-2N1R  | on         | on            | Natural DSD + "seeding" particle($R_d = 0.1\ \mu m$, $R = 4\ \mu m$, $N = 20\ cm^{-3}$) |
|                | blueRun Seed-1N2R | on      | on            | Natural DSD + "seeding" particle ($R_d = 1\ \mu m$, $R = 8\ \mu m$, $N = 10\ cm^{-3}$) |

Run CTL is the control run. Only one condition is changed in each of the other five cases. Runs CTL, NoTurb, and NoSolu use the same initial DSD from the parcel model and are referred to as the "natural" cases. Turbulence and solute effect are switched off in Run NoTurb and Run NoSolu, respectively, to gauge the effects of turbulence and CCN hygroscopicity on the DSD. When turbulence is switched off, the background velocity fluctuation is set to $0\ m\ s^{-1}$. Therefore, particle motion

is only affected by the mean updraft and gravitational settling, and the supersaturation fluctuation is only induced by droplet condensation and evaporation. When the solute term is switched off, i.e., $\kappa = 0$, droplets consist of only pure water. Runs Seed-1N1R, Seed-2N1R, and Seed-1N2R are referred to as "seeded" cases because an extra number of monodisperse aerosols are introduced near the cloud base (at the beginning of DNS). Two seeding sizes and two number concentrations are considered, as described in Table 2. Different than the traditional cloud seeding, the same hygroscopicity of $\kappa = 0.47$ is assumed for both the natural aerosols and the seeding particles. In Run Seed-1N1R, we introduce seeding particles of dry radius $R_d = 0.1\ \mu m$, wet radius $R = 4\ \mu m$, and number concentration $N = 10\ cm^{-3}$. We double the seeding particle number concentration in Run Seed-2N1R. In Run Seed-1N2R, the dry size of the seeding particles increases tenfold and the wet size doubled, relative to Run Seed-1N1R (see Table 2). It should be pointed out that the dissipation rate in cumulus clouds tends to increase with height (Seifert et al., 2010). For simplicity, the eddy dissipation rate ($\epsilon$) for all the turbulent cases is set statistically stationary ($\epsilon = 500\ cm^2\ s^{-3}$). The advantage of this idealized, simplified treatment is that the effect of turbulence can be easily separated from aerosol effects. A dissipation rate of $500\ cm^2\ s^{-3}$ represents a strongly turbulent environment in cumulus clouds to examine the upper-bound of turbulent effects on the DSD evolution.

## 3 Results

### 3.1 Natural cases

We first compare the results of the natural cases (Runs CTL, NoTurb, and NoSolu) to examine the effect of turbulence and hygroscopicity (solute) on the droplet evolution. Fig. 3 shows that including solute and turbulence effectively broaden the DSD at different times. With droplets containing no solute in Run NoSolu, the DSD broadening is suppressed within the first six minutes. However, the tail evolution quickly catches up and converges to that in Run CTL afterwards. Meanwhile, switching off turbulence in Run NoTurb suppresses the DSD broadening at a later time (Fig. 3). The tail of the spectrum in Run CTL and Run NoTurb stays similar in the first two minutes and starts to differ by a large amount afterwards.

Turbulence effects on the DSD broadening is minor before T=6 min (Fig. 4(a-b). Both Run CTL and Run NoTurb produce a similar number of droplets over $25\ \mu m$ at T=6 min. The majority of this size group is grown from the ultra giant aerosol with an initial dry and wet size of $R_d = 4.9\ \mu m$ and $R = 16\ \mu m$. They grow rapidly to $25\ \mu m$ by condensation within the first two minutes in Run CTL and Run NoTurb. However, droplets can hardly reach beyond $30\ \mu m$ solely by condensation (Fig. 4 (d-e)). The tail over $30\ \mu m$ is mainly formed by the subsequent collision-coalescence process. Once droplets are over $25\ \mu m$, the gravitational collection becomes effective, leading to a similar DSD tail with or without turbulence. However, gravitational collection of droplets below $25\ \mu m$ in Run NoTurb is ineffective to sustain the formation of large droplets. After T=6 min, the tail of DSD in Run NoTurb becomes quasi-stationary for droplets over $20\ \mu m$ (red and blue histograms in Fig. 4(b)) due to negligible gravitational collisions. This can be illustrated by a negligible collision frequency in Run NoTurb in Fig. 6(e). In contrast, a substantial number of droplets $> 20\ \mu m$ are constantly formed in Run CTL after T=2 min through rapid turbulent collisions. Comparing to collision frequency in Run NoTurb (Fig. 5 (b)), turbulence substantially enhances the collisional growth of droplets of $R < 20\ \mu m$. The total collisions in turbulent cases increase by a factor of 20. It is also found that the

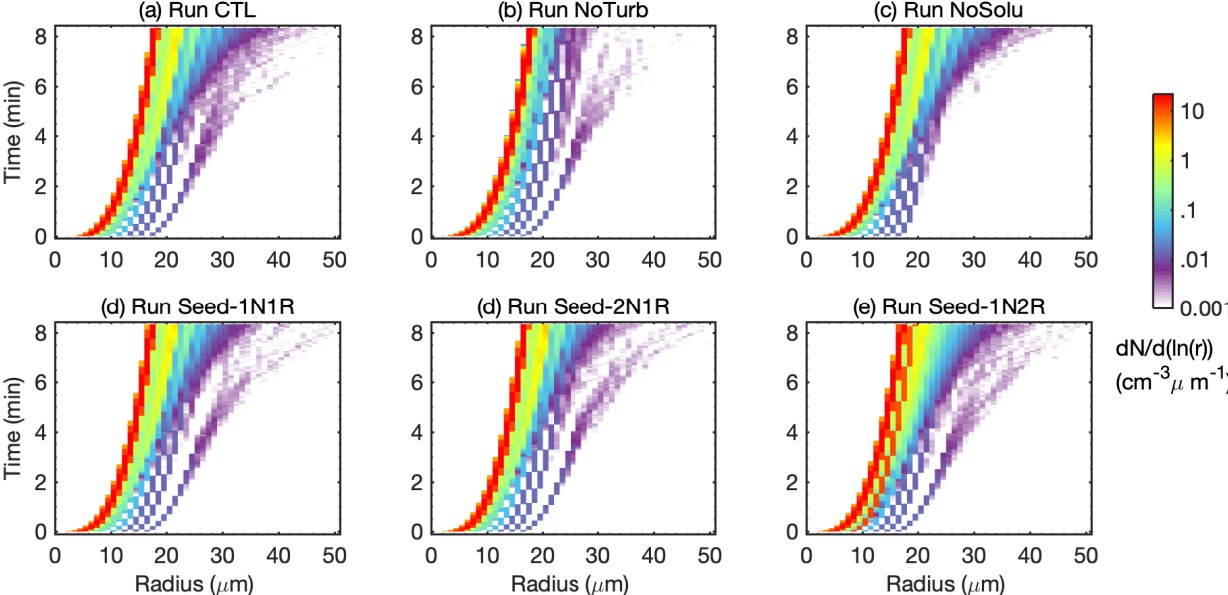

**Figure 3.** Time evolution of the droplet size distribution in the condensation-collision set of experiments. The droplet number concentration ($cm^{-3}$) is indicated by colors with its value shown in the color bar. The configuration of each experiment is listed in Table 2.

turbulent enhancement of collisions is strongest among droplet pairs of similar sizes, i.e., with a radius ratio of $r/R > 0.8$. Similar-sized collisions increase by nearly a factor of 50 in turbulent cases, contributing to over 80% of the total collisions as opposed to 34% in Run NoTurb. This is because a non-turbulent environment does not favor similar-sized collisions due to a similar droplet settling speed. Turbulence, on the one hand, increases the relative motion between droplets and on the other hand, induces a stronger clustering of similar-sized droplets. The two effects jointly strengthen the similar-sized collisions. The turbulent enhancement on similar-sized collisions is then amplified by the condensational process. Chen et al. (2018b) also demonstrated that as the condensation process reduces the DSD width and generates more similar-sized droplets, turbulence enhances the similar-sized collision and thus broadens the DSD.

Even though turbulence intensifies the collisional growth, the modulation on the droplet condensation is found insignificant. The DSDs in Run CTL and NoTurb in the condensation-only set are nearly identical (Fig. 4 (d-e)). This is because the supersaturation fluctuations are weak in an adiabatic core region. Vaillancourt et al. (2002) found that in a quasi-adiabatic environment both particle sedimentation and short-lived turbulent coherent structure reduce the supersaturation fluctuation and decrease the time that droplets are exposed to these fluctuations. We expect that the turbulent-induced condensational broadening is more significant in the cloud edge where entrainment mixing induces large variation in supersaturation fluctuations.

When solute effect is absent in Run NoSolu, droplets can hardly reach beyond 30 $\mu m$ before T=6 min (Fig. 4 (c)) because of a lack of ultra-giant aerosols ($R_d > 4\ \mu m$). Embryonic drizzle drops at the early-stage (T<6 min) are formed from the fast growth of the ultra-giant aerosols as seen in both Run CTL and Run NoTurb. No significant change is found in the mean droplet

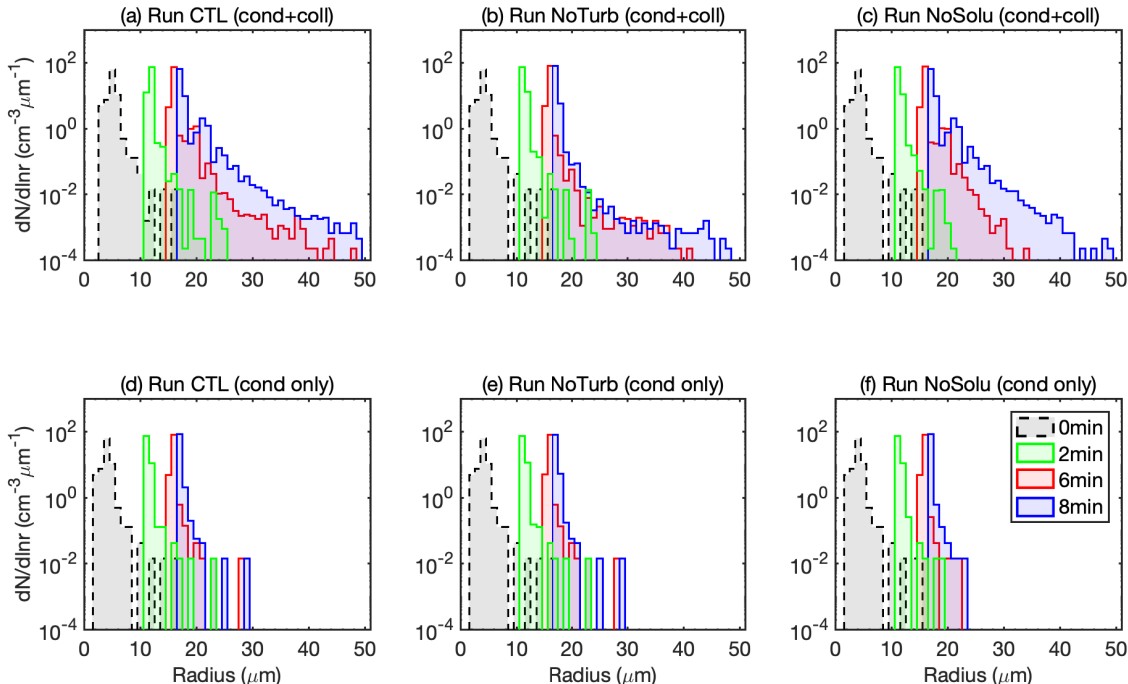

**Figure 4.** Droplet size distributions (DSDs) at $T = 0\ min$ (grey), $T = 2$ min (green), $T = 6\ min$ (red), and $T = 8\ min$ (blue) of the natural cases (Runs CTL, NoTurb, and NoSolu). The upper panel (a-c) is the DSD in the condensation-collision set of experiments, and the lower panel (d-e) is the DSD in the condensation-only set of experiments.

radius and the relative dispersion between Run CTL and Run NoSolu (Fig. 7(d)). Only a slightly lower collision frequency in the droplet size group of $R > 20\ \mu m$ results from a lack of ultra-giant aerosols (see the green histograms in Fig. 5). This implies that the solute effect on droplet condensation in DSD broadening is small for aerosols below $R_d < 4\ \mu m$. The ultra giant aerosols ($R_d = 4.9\ \mu m$ in this study), due to their scarcity, have a negligible contribution in shifting the mean radius and relative dispersion (Fig. 7). As shown in Fig. 3(c), an efficient broadening is triggered at T= 6 min, resulting in a similar DSD as in Run CTL at the end of the simulation. It is shown that droplets between $20 - 30\ \mu m$ are produced through turbulent collisions by the end of T = 6 min (Fig. 4(c)), causing a boost in collisions of droplets over $20\ \mu m$ (Fig. 6(d)).

The time evolution of collision frequency in Fig. 6 shows that all five turbulent cases show a similar trend in total collisional frequency, even though the trend at the four size groups varies. The non-turbulent gravitational collection process is very weak with the collision frequency lowered by at least one order of magnitude in Run NoTurb. Still, a slightly higher droplet number concentration at $R > 40\ \mu m$ is observed in Run CTL and Run NoTurb than in Run NoSolu, because of the presence of ultra giant aerosols. At the same time, the collision frequency of the four size groups in Run CTL and Run NoSolu are

almost identical. Even though the ultra giant aerosols are important in forming early drizzle embryos, due to a low number concentration, they do not sustain an efficient collectional process.

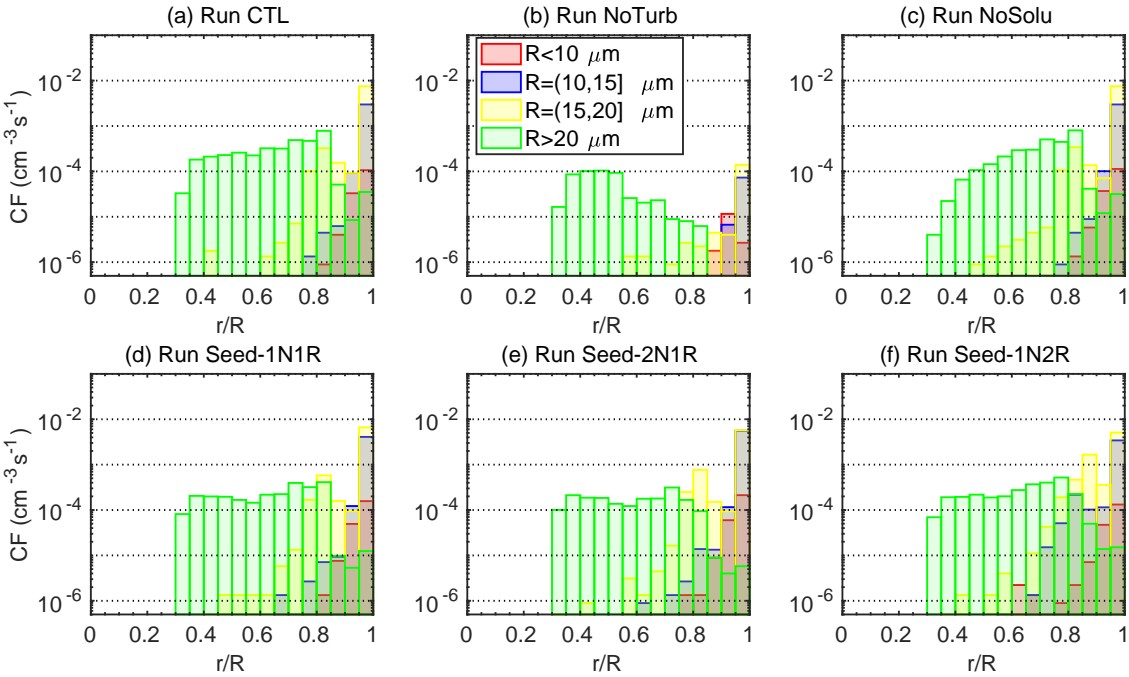

**Figure 5.** Collision frequency (CF) varying with $r/R$ in the condensation-collision set of experiments. $r/R$ is the radius ratio between the small droplet and the large droplet of collided droplet pairs. The droplet pairs are divided into four size groups by the big droplet radius, R, shown in the legend.

The relative dispersion, defined as the ratio between the standard deviation of the DSD and the mean droplet radius, is an indicator of the width of the DSD. The values among the six cases at the end of the simulation range from $0.01 - 0.1$, which is highly consistent with the theoretical study by Liu et al. (2006b, Fig. 1) for an aerosol number concentration close to $100\ cm^{-3}$. The dashed lines in Fig. 7 (c) demonstrate that condensational growth narrows the DSD and decreases the relative dispersion throughout the simulation in the condensation-only set. Droplet growth in the first two minutes is prevailed by condensation,

as the relative dispersion in the condensation-collision set of experiments well overlaps with that in the condensation-only set. After T = 2 min the relative dispersion in the condensation-collision set and the condensation-only set starts to deviate from one another. This is mainly due to two factors: 1) the condensation narrowing slows down as droplets get larger and supersaturation gets lower; 2) the collision rate increases with the increasing droplet mean radius and thus leads to a higher collision rate to strengthen the DSD broadening. In Run NoTurb, the collision rate stays the lowest of all cases throughout the simulation (Fig.

6 (e)), leading to the smallest relative dispersion of all the six cases.

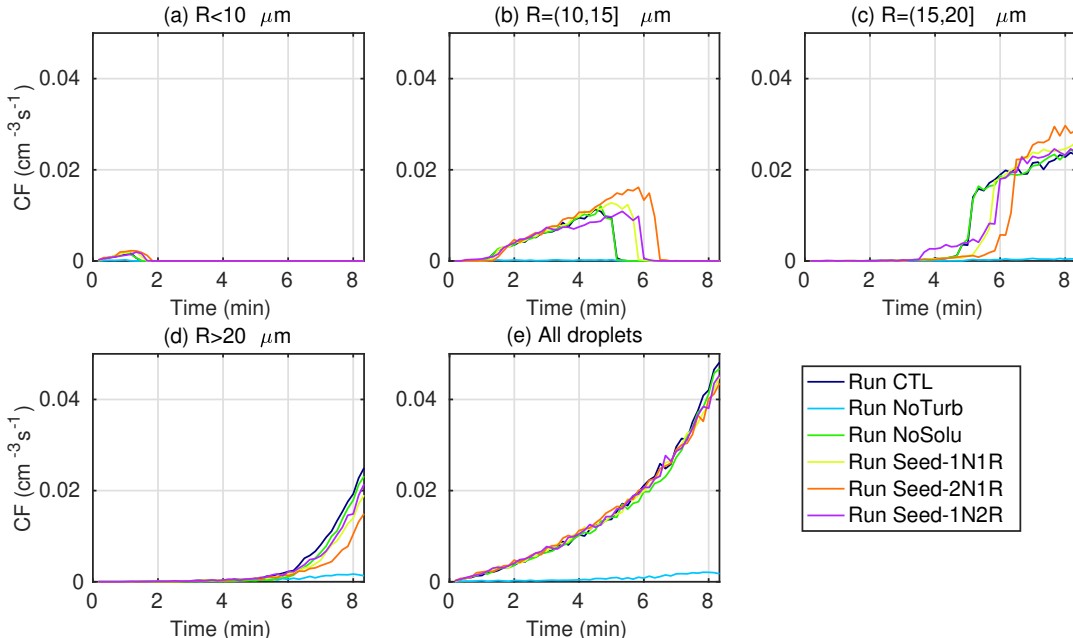

**Figure 6.** (a-d) Time evolution of collision frequency for droplet pairs of four different size groups mentioned in Fig. 5. (e) Time evolution of collision frequency for all droplet pairs.

Despite that DSDs differ among the six cases, the modulation of the bulk condensation by both turbulence and aerosol is negligible, as supported by an almost identical LWC of the six cases (Fig. 7(a)). This is because the fall out mass of drizzle drops of $R > 50~\mu m$ before T=500 s is negligible, and the bulk LWC of the six cases is approximately adiabatic. Turbulence and aerosols redistribute water mass among different droplet sizes by modifying the condensational and collisional growth of individual droplets, thus shifting the droplet statistics such as the mean radius and relative dispersion, and eventually alters the autoconversion rate (Fig. 7(f)). The autoconversion rate here is defined as the mass transfer rate from droplet smaller than $R = 30~\mu m$ to droplet larger than $30~\mu m$. It is also found that even though Run NoTurb produces the second largest mean radius, the autoconversion rate stays the lowest, accompanied by the smallest relative-dispersion. Therefore, properties such as the shape of the DSD and relative dispersion are more relevant to autoconversion than the LWC. The traditional autoconversion parameterizations such as the Kessler-type parameterization (Kessler, 1969; Liu and Daum, 2004) and the Sundqvist-type parameterizations (Sundqvist, 1978; Liu et al., 2006a) customarily use a threshold function based on the mean radius and/or the LWC. It is suggested that autoconversion rate is also influenced by various other parameters (see Noh et al., 2018, and references therein). The present study demonstrates that both parameters, in particular, the LWC cannot properly capture the trend of the autoconversion. The autoconversion rate by Berry and Reinhardt (1974), and its modified versions which include both the mean droplet size and dispersion parameter, is conceptually better than the Kessler-type schemes. Our results thus agree with Gilmore and Straka (2008) which found that the scheme of Berry and Reinhardt (1974) is more sophisticated and requires less tuning to match the observed onset of rain and proportions of cloud and rain. They also found that the growth

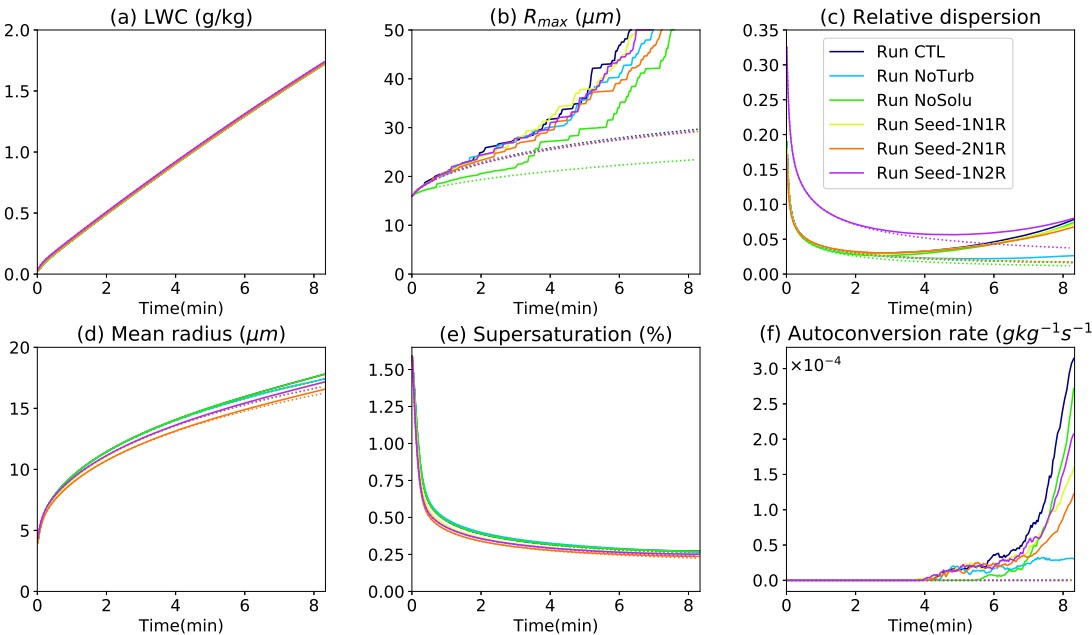

**Figure 7.** The temporal variation of bulk (a) liquid water content (LWC), (b) maximum droplet radius ($R_{max}$) , (c) relative dispersion, (d) droplet mean radius, (e) supersaturation ratio, and (f) autoconversion rate in the condensation-collision set of experiments (solid lines) and in the condensation-only set of experiments (dotted lines). The relative dispersion is defined as the standard deviation of the droplet radius divided by the mean radius. The autoconversion rate here is defined as the mass transfer rate from droplets smaller than $R = 30$ $\mu m$ to droplets larger than 30 $\mu m$. The droplets over 50 $\mu m$ are treated as fall-outs and removed from the domain. Thus (b) only shows a maximum droplet size at 50 $\mu m$.

rate of rain mass and number concentration are highly sensitive to the shape and dispersion parameters. Additionally, it is worth noting that turbulence modifies the collision rate and thus shifts the DSD shape and relative dispersion. Therefore, a 290 turbulence-dependent relative-dispersion parameter is needed in developing the autoconversion scheme.

## 3.2 Seeded cases

Seeding reduces the mean droplet radius due to higher competition for water vapor among individual droplets (Fig. 7 (d)). Therefore seeding slows down the autoconversion process. Nevertheless, the LWC is not affected by seeding (Fig. 7)a)), which again indicates that the LWC is not a well-related quantity to autoconversion in this case.

When investigating the relative importance of aerosol and turbulence to droplet growth, it is found that the modulation of droplet mean radius by seeding particles is larger than the modulation by collision-coalescence. In Fig. 7 (d), the difference between seeded and unseeded cases exceeds the difference between the condensation-only set (dotted lines) and condensation-

collision set (solid lines) of each case. Regardless, turbulent collision-coalescence yields large droplets over 30 $\mu m$ and increases the width of the DSD. The total collision rate is heavily determined by the turbulence level and mildly affected by seeding or CCN hygroscopicity (Fig. 6(e)). Besides, the change in $R_{max}$ and relative dispersion due to collisions exceeds that from changing the aerosol condition. As condensational growth can hardly produce droplets over 30 $\mu m$, turbulent enhancement of collision is determinant in the mass conversion from small droplets to drizzle embryos. Meanwhile, seeding increases the competition for water vapor among droplets and reduces the mean droplet size, leading to more collisions of small droplets and fewer collisions of large droplets (Fig. 6(a-d)). Specifically, by doubling the seeding particle number in Run Seed-2N1R, the condensational growth of small droplets is further inhibited due to a higher competition of water vapor, resulting in more small droplets. Increasing the size of seeding particles in Run Seed-1N2R buffers the above-mentioned inhibition effect caused by increasing aerosol number concentration. The resulting autoconversion rate ordering is Run CTL > Run Seed-1N2R > Run Seed-1N1R> Run Seed-2N1R.

Finally, aerosol hygroscopicity is key to the onset time of autoconversion. All five solute-containing cases see a similar onset time around T= 4 min. Removing the solute (hygroscopic material) in Run NoSolu delays the onset of autoconversion by about 1.5 min (green line Fig. 7 (f)). Nevertheless, after T = 6-7 min, the autoconversion rate in Run NoSolu exceeds all seeded cases. First, solute (CCN hygroscopicity) has a negligible effect on the growth of small aerosols, as the size distribution of small droplets in Run CTL and Run NoSolu remain almost identical. This is substantiated by the almost identical collision frequency of droplets below 20 $\mu m$ of the two cases (Fig. 6 (a-c)). Second, seeding reduces the mean radius of the droplets. This leads to a reduction in collisions for droplets over 20 $\mu m$ (Fig. 6(d)) and subsequently decelerates the autoconversion process. The above findings imply that increasing the aerosol size (ultra-giant aerosol) shortens the lifetime of the clouds through a fast onset of rain. And increasing the number of aerosols decelerates the rain process.

## 4 Summary and discussion

This paper investigates the effects of turbulence and aerosol properties (hygroscopicity, number concentration, and size) on the microphysics during early cloud and rain development. A parcel-DNS hybrid modeling framework is developed. The parcel model is used to generate the initial size distribution of activated aerosols, and the DNS model calculates the subsequent growth of those activated aerosols affected by both the microscopic (turbulent fluctuation) and the macroscopic (bulk) environment. By using this economical modeling framework, continuous particle growth from sub-cloud aerosols to cloud droplets is accurately represented.

Overall, ultra-giant aerosols in the natural cases quickly form the drizzle embryo and thus determine the onset time of autoconversion. However, they only form a few big raindrops due to their scarcity, which has little impact on the level of autoconversion. Turbulence enhances the collision frequency by more than one order of magnitude and determines the level of autoconversion. Specifically, turbulence enhances the collisions among similar-sized droplets that are less likely to happen in a non-turbulent environment, effectively broadening the DSD. Therefore, the autoconversion in a turbulent environment is significantly greater than in a non-turbulent environment. It is also found that seeding (increasing aerosol number and

size) modifies the level of autoconversion. On the one hand, increasing the aerosol number reduces the mean radius due to stronger competition for water vapor, and therefore slows down the autoconversion. On the other hand, increasing the seeding size can buffer such negative feedback. However, the seeding particles in this study only cover a limited range of dry radius ($R = 0.1,\ 1\ \mu m$) and number concentration ($N = 10,\ 20\ cm^{-3}$, corresponding to $10 - 20\%$ increase in the total number concentration). Conditions with more ultra-giant aerosols ($R \gg 1\ \mu m$), lower aerosol concentrations ($N \ll 100\ cm^{-3}$), or highly polluted environment ($N \gg 100\ cm^{-3}$) will be of interest to further assess the relative importance of aerosols and turbulence. It is argued that predicting the rain onset time requires accurate information and representation of ultra-giant aerosols. And an accurate autoconversion scheme requires a well-quantified turbulent collisions kernel.

Even though the autoconversion rate differs among the six cases, it is found that the bulk variables such as LWC, mean radius, and supersaturation are not sensitive to turbulence level and aerosol conditions. In this case the LWC and mean droplet radius, which are key parameters in Kessler-type or Sundqvist-type autoconversion parameterizations, are not well-related quantities to autoconversion rate, and information of turbulence intensity and aerosols are essential to determine the autoconversion rate. It is argued that these bulk variables are mainly affected by the updraft speed which is held the same among the six cases. Sensitivity studies are needed in the future to investigate the effect of the LWC on the autoconversion rate due to a change in the updraft.

Cloud models are sensitive to microphysics schemes, and the autoconversion parameterization is one of the main sources of uncertainty in the representation of warm clouds and rain with few observations to verify against. The large uncertainty may be ascribed to the decoupling of microphysics from subgrid-scale turbulence and a lack of aerosol information in the parameterization. Therefore, the aerosol effect evaluated by the models should be cautiously interpreted. The hybrid parcel-DNS model can be used for verifying the autoconversion rate affected by turbulence and aerosols at the sub-grid scale of large-eddy simulation (LES).

Despite a good number of improvements made, the current modeling framework still presents the following shortcomings: for simplicity, the same hygroscopic parameter ($\kappa = 0.47$) is assumed among the natural aerosols and the seeding particles. Besides, seeding is initialized $40\ m$ above the cloud base while traditional hygroscopic seeding introduces particles around $100 - 300\ m$ below the cloud base. This treatment might affect the model results as seeding below the cloud base influences the activation and growth of the background aerosols and thus modifies the DSD at the cloud base (Cooper et al., 1997).

Our idealized simulations focus on the cloud adiabatic core region and therefore exclude entrainment mixing which is highly active near the cloud edge. Activation of laterally entrained aerosols might occur in cumulus clouds outside the adiabatic core (Hoffmann et al., 2015; Slawinska et al., 2012). The newly activated aerosols might lead to a further broadening of the DSD (e.g., Lasher-Trapp et al., 2005). In addition, the in-cloud mixing at a much larger scale than the DNS domain transports and mixes both the air and droplets from different parts of the cloud including the cloud edge, leading to a highly perturbed Lagrangian history of supersaturation experienced by droplets (Grabowski and Abade, 2017, "eddy hopping effect"). On the other hand, larger turbulent eddies can generate higher supersaturation fluctuations due to a higher variation in a vertical motion and thus may both affect the aerosol activation and broaden the DSD. Traditional DNS which is confined to a relatively small domain size (<1 m), and the impact of supersaturation fluctuations is significantly restricted. Methods such as an up-scaled

DNS with superdroplets (e.g., Thomas et al., 2020) or representing the large-scale mixing with an external forcing on the thermodynamic fields (Paoli and Shariff, 2009) can be used for studying the impact of turbulent scales on the supersaturation fluctuations and thus on the condensational broadening of DSD. In conclusion, the relative importance of entrainment, eddy hopping effect, small-scale turbulence and aerosols requires further investigation.

This study proposes the first DNS model framework for scrutinizing the microphysical impact of cloud seeding and presents the first results of such a model. Full DNS modeling from below the cloud base will be the next step to include the effect of turbulence on aerosol activation. Additionally, more realistic scenarios resembling actual hygroscopic seeding conditions, such as utilizing multi-disperse size distributions, different hygroscopicity parameters, and seeding below the cloud base will be designed in the future development and deployment of this framework.

*Code and data availability.*  The data produced by the Direct Numerical Simulation (DNS) model and parcel model can be accessed in the Harvard Dataverse repository (Chen et al., 2019, doi:10.7910/DVN/HBIKKV). The parcel model and DNS model used to produce the dataset are available upon request.

*Author contributions.*  This study was co-designed by Sisi Chen, Lulin Xue, and M.K. Yau. Sisi Chen conducted the model simulation, did the data analysis, and wrote the manuscript. Lulin Xue and M.K. Yau provided advice and discussions on the model results and revised the
manuscript.

*Competing interests.*  The authors declare that they have no conflict of interest.

*Acknowledgements.*  We thank the two anonymous reviewers for their invaluable comments. This work is supported by the Advanced Study Program at National Center for Atmospheric Research, sponsored by the National Science Foundation under Cooperative Agreement No. 1852977. Part of this work is supported by the National Center of Meteorology, Abu Dhabi, UAE under the UAE Research Program for Rain
Enhancement Science. We would like to acknowledge high-performance computing (HPC) support from Cheyenne, Graham, and Cedar. HPC resources at Cheyenne (doi:10.5065/D6RX99HX) is provided by NCAR's Computational and Information Systems Laboratory and sponsored by the National Science Foundation. HPC resources at Graham and Cedar are provided by Compute Canada (www.computecanada.ca).

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
