# Peer review of "Impact of aerosols and turbulence on cloud droplet growth: An in-cloud seeding case study using a parcel-DNS approach"

_Atmospheric Chemistry and Physics, 2019_

## Referee Comment (RC1) · Anonymous Referee #1 · 19 Nov 2019

**Review of "Impact of hygroscopic CCN and turbulence on cloud droplet growth: A parcel-DNS approach" by Chen et al. (acp-2019-886)**

The submitted study investigates the effects of turbulence and aerosols (number and hygroscopicity) on cloud droplet growth by condensation and collection. By applying a unique combination of direct numerical simulation (DNS) with a parcel model, the authors show that turbulence increases droplet collection, while aerosol hygroscopicity increases spectral broadening, in agreement with several previous studies. Furthermore, the authors show that additional aerosols lead to a reduced mean droplet radius as expected. However, the simultaneously reduced liquid water content — in the presented magnitude and within the applied framework — remains a riddle to the reviewer.

The general topic of this study is of interest and the applied modeling framework is expected to lead to new insights. However, (potentially false) conclusions are based on an inadequate analysis of the modeling results. Additionally, the manuscript lacks appropriate consideration of previously published literature and requires some clarification in writing. While I believe that these weaknesses can be remedied, there are too many and too severe concerns at this stage. Therefore, I must suggest rejecting the manuscript from publication in Atmospheric Chemistry and Physics in the current form. I will support my suggestion in more detail below.

**Major Comments**
*Previous literature*
The broadening of droplet size distributions has been studied intensively in the past, including the effects of aerosol hygroscopicity and turbulence (supersaturation fluctuations). Of course, the representation of these processes was highly idealized in the past, but the conclusions are not different from the submitted manuscript. For instance, Srivastava (1991) showed that the consideration of curvature in the diffusional growth equation is essential for the spectral broadening of a droplet size distribution in a lifted parcel. Furthermore, Korolev (1995) showed that curvature and solute terms lead to irreversible broadening if supersaturation fluctuations are applied. Further analysis has been carried out by Çelik and Marwitz (1999). Integrating these very simple, even analytical investigations into the highly complex framework of the submitted study will increase physical understanding and insight.

*Analysis*
Many conclusions are based on Fig. 7. While the authors are candid about the fact the mean radius (panel a), the dispersion (panel b), the maximum radius (panel c), and the radar reflectivity (panel e) are calculated from droplet size distribution, I believe that this is also the case for the liquid water content (LWC) in panel d. Since the parcel is ascending with a constant velocity, the LWC is expected to increase linearly with time: $LWC = w\,t\,\Gamma_W$, with $\Gamma_W \approx 2.0 \times 10^{-6}\ \mathrm{kg\,m^{-4}}$. While the LWC follows this linear relationship for the first three minutes, it deviates significantly afterward. Of course, the LWC may deviate from this adiabatic behavior if the transfer of water vapor into the liquid phase is kinetically limited. However, this effect can be excluded since its magnitude decreases with droplet size and hence time. One reasonable explanation for this unphysical behavior is that the authors diagnosed the LWC from the droplet size distribution (as also suggested by the similarity to the mean radius in panel a) or made another error in the calculation of the LWC. Accordingly, any conclusions on LWC differences between the individual model runs are potentially false (e.g., ll. 9 – 10, 231 – 240, 265 – 268). Therefore, I strongly suggest repeating all simulations of this study, and to diagnose the LWC directly from the simulated droplets. I further suggest doing the same for the mean radius, dispersion, and maximum radius.

*Limitations of the Modeling framework*
The authors state that the applied modeling framework promotes a more realistic assessment of cloud microphysical processes during the early development of a cloud. And I agree. However, there are certain limitations and restrictions inherent to this approach that need to be addressed. In fact,

they might restrict the scope of the manuscript's conclusions significantly. First, the ascent of the air parcel is adiabatic. In cumulus clouds, entrainment dilutes the cloud constantly, reducing the liquid water content, while the activation of newly entrained aerosols might lead to a further broadening of the droplet size distribution (e.g., Lasher-Trapp et al. 2005). Second, the kinetic energy dissipation rate is assumed to be constant throughout the ascent, using a relatively high value only typical for the top of shallow cumulus clouds. This is a crude simplification since turbulence tends to increase with height (e.g., Seifert et al. 2010). Fourth, the DNS domain is relatively small. Restricting vertical motions to 16.5 cm, the impact of supersaturation fluctuations is significantly restricted (e.g., Abade et al. 2018). Fifth, the range of investigated aerosol concentrations is relatively narrow. Especially the investigation of even lower aerosol concentrations might be of interest to assess the relative importance of aerosol hygroscopicity and turbulence even further.

**Minor Comments**

Ll. 3, 69: The term "aerosol loading" feels ambiguous here. I believe you want to investigate the effects resulting from changes in the number of CCN.

L. 8: Inhibiting is a very strong word here. As long as the smaller droplets are activated and the domain is supersaturated, their growth might be reduced but not inhibited.

L. 9: "[…] seeding reduces the LWC […]". See major comment and revise accordingly.

Ll. 9 ff.: The term *effective radius* is used interchangeably with *mean radius* within the entire manuscript. While the *mean volume radius* is usually close to the effective radius, this is not necessarily the case for the *arithmetic mean radius*. Please clarify.

Ll. 35 – 36: The comment regarding the effects of cloud chamber walls is appreciated. However, I feel that this relatively specific effect might demand a reference, e.g., Thomas et al. (2019).

Ll. 43 – 44, 89: Why is it necessary to divide the model into the two steps using a standard parcel model first and subsequently a DNS? I assume it is straightforward to consider the activation of aerosols within the DNS framework, similar to the parcel model. In fact, considering turbulent supersaturation fluctuations during activation might have important implications for the estimated number of cloud droplets (Abade et al. 2018). Overall, this could result in a more consistent modeling approach. And based on the data provided in the manuscript, one could easily estimate that the total computing increases by less than 30 % when the DNS is used for the entire ascent. I consider this increase as acceptable.

L. 84: Add the height of the maximum supersaturation to the text.

Caption to Fig. 2: Is it really the relative altitude? It is still the (absolute) height above cloud base. A *relative* altitude is usually divided by another quantity for normalization.

Ll. 92 – 93: The fact that droplets smaller than 10 µm in radius collide rarely is a well-known fact, and can be obtained from any text book on cloud physics.

L. 94: I believe that the hygroscopicity parameter is represented by κ in the Eq. (1) and (2). Am I right? I suggest stating this explicitly.

Ll. 123 – 124: How do you distinguish between activated and unactivated aerosols? Large aerosols do not need to be activated (i.e., be larger than the critical radius) to behave like a droplet.

Ll. 126 – 129: Does this manipulation also change the liquid water content in the DNS simulation?

Ll. 136 – 139, Eq. (2): Why don't you use the same diffusional growth equation for the parcel and the DNS model?

Table 1: I suggest using simpler names for the model runs, indicating directly the difference to the control run. E.g., "run NoTurb" instead of "run B".

Ll. 165 – 166: This is misleading since the particles are still lifted with a mean updraft of 2 m s$^{-1}$.

L. 166: What is meant by "turbulent advection of the supersaturation fluctuation"? Is there no turbulent mixing in the DNS domain, only molecular diffusion, when "turbulence is switched off"? This might overestimate the effects of supersaturation fluctuations caused by processes other than turbulence, e.g., the faster depletion of supersaturation in the vicinity of a large droplet.

L. 174: A kinetic energy dissipation rate of 500 $cm^2$ $s^{-3}$ at cloud base is too high. Typically, the dissipation rate increases with distance to cloud base, and a value of 500 $cm^2$ $s^{-3}$ might only be representative for the top of a shallow cumulus cloud.

Sec. 3: I suggest introducing subsections to increase clarity of this section.

L. 177: You state that the parcel rises up to 1.2 km above cloud base. This ascent lasts 600 s = 10 min, assuming an updraft of 2 m $s^{-1}$. Accordingly, the analysis after 6 min does not constitute the end of the simulation. Moreover, it is also not clear if this point in time is considered to be after the start of the DNS or after the start of the parcel model.

L. 184: "Condensational growth after 1 min becomes extremely slow […]". How can we distinguish between condensational growth and collision-coalescence in Fig. 5b?

L. 187, Fig. 7b: The displayed dispersion values are much smaller than the values observed in real clouds. In fact, they are about one order in magnitude smaller. You should comment on this difference and name reasons.

Ll.188 – 190: What is meant by "multi-modal feature"? For me, the droplet spectra seem to be almost monomodal. There might be a second mode developing between 20 and 25 μm, but it is not very distinct.

Fig. 3e: After 6 min, small (radius < 1 μm) droplets seem to appear in the droplet size distribution. Where do they come from?

Ll. 210 ff.: By CCN case you mean the case with solute effects (but no turbulence) or the control case?

Ll. 221 – 223: In what process are the hygroscopic CCN more effective in the first few minutes?

L. 226: Define large droplets by their size range.

Ll. 234 – 238: It is impossible that a higher droplet concentration (as a result of the seeding) results in a lower liquid water content. In fact, in a rising parcel without (interactive) entrainment, one would assume a higher liquid water content due to the accelerated depletion of water vapor. See major comments above.

L. 243: A "flatter and broader" droplet size distribution sounds tautological. Since you do not change the number of CCN significantly, a broader distribution needs to be flatter. Or does "flat" refer to a another property of the distribution that I miss?

Ll. 231 – 247: All seeding experiments tend to address the effect of additional CCN. Those will always decrease the condensational growth of individual droplets since the water is distributed on a larger number of droplets. The more interesting case would be a reduction in the number of CCN.

Fig. 7: The colors are barely distinguishable. Please change them. Why do you show the radar reflectivity (panel e), which is not discussed in the text?

Ll. 265 – 268: This conclusion is based on inadequate analysis and is not true as outlined in the major comments.

Ll. 270 – 272: This claim is only true because of the limited range of CCN concentrations tested in this study. For the analyzed cloud, it will probably always rain if the CCN concentration is reduced to 10 $cm^{-3}$ and it will probably never rain if it is increased to 1000 $cm^{-3}$, irrespective of turbulence or aerosol hygroscopicity.

**Technical Comments**

Language: While the language is understandable, several smaller mistakes slow down the reading process.

L. 3: Define DNS in the abstract.

L. 39: Remove "Lagrangian-tracking".

L. 43: DNS has already been defined in l. 36.

L. 56: It is odd citing Chen et al. (2018b) before citing Chen et al. (2018a).

L. 61: I suggest adding "subsaturated" before downdrafts.

L. 81 ff.: Units are usually stated in upright characters and separated by a thin space (\, in LaTeX) from the numerical value.

L. 107, several other occasions: I suggest using exponents for units (m $s^{-1}$ instead of m/s), consistent with the notation used throughout the manuscript.

L. 200: Why do you introduce the abbreviation BR74, which is used only once in the manuscript?

Fig. 7d: It is g/kg, and not g/Kg.

**References**

Abade, G. C., Grabowski, W. W., and Pawlowska, H. (2018). Broadening of cloud droplet spectra through eddy hopping: Turbulent entraining parcel simulations. *Journal of the Atmospheric Sciences*, *75*(10), 3365-3379.

Çelik, F., and Marwitz, J. D. (1999). Droplet spectra broadening by ripening process. Part I: Roles of curvature and salinity of cloud droplets. *Journal of the atmospheric sciences*, *56*(17), 3091-3105.

Korolev, A. V. (1995). The influence of supersaturation fluctuations on droplet size spectra formation. *Journal of the atmospheric sciences*, *52*(20), 3620-3634.

Lasher-trapp, S. G., Cooper, W. A., and Blyth, A. M. (2005). Broadening of droplet size distributions from entrainment and mixing in a cumulus cloud. *Quarterly Journal of the Royal Meteorological Society: A journal of the atmospheric sciences, applied meteorology and physical oceanography*, *131*(605), 195-220.

Seifert, A., Nuijens, L., and Stevens, B. (2010). Turbulence effects on warm-rain autoconversion in precipitating shallow convection. *Quarterly Journal of the Royal Meteorological Society*, *136*(652), 1753-1762.

Srivastava, R. C. (1991). Growth of cloud drops by condensation: Effect of surface tension on the dispersion of drop sizes. *Journal of the Atmospheric Sciences*, *48*(13), 1596-1599.

Thomas, S., Ovchinnikov, M., Yang, F., van der Voort, D., Cantrell, W., Krueger, S. K., and Shaw, R. A. (2019). Scaling of an atmospheric model to simulate turbulence and cloud microphysics in the Pi Chamber. *Journal of Advances in Modeling Earth Systems*, *11*(7), 1981-1994.

---

## Referee Comment (RC2) · Anonymous Referee #2 · 25 Nov 2019

The submitted paper describes a set of simulations of evolution of cloud droplet spectrum within non-entraining adiabatic air parcel. The focus of the study is the impact of in-cloud seeding of the cloud with monodisperse droplets on the final cloud and drizzle drop spectra. The simulations include representation of turbulent inhomogeneities using a Direct Numerical Simulation (DNS) approach, with the DNS box representing the adiabatic parcel (with a volume of ca. 4.5 cubic litres). The simulation scenario is a 1500 m parcel ascent with a constant vertical velocity of 2 m/s with:

i) representation of spatial variations in heat and moisture disabled until reaching the level of maximal saturation (i.e., after ca. 350 m of ascent) - this phase assumes homogeneous ambient conditions;

ii) seeding with droplets (of 4 or 8 $\mu m$ in radius) happening instantaneously at the level

of maximal saturation;
iii) further ascent for next ca. 1150 metres with DNS-resolved effects of turbulent inhomogeneities as well as with simulation of collisional growth.

Technically, point (i) is realised in the presented simulations by using a parcel model with Lagrangian-in-radius size spectrum evolution below the level of maximal saturation, and then initialising the DNS simulation with particle population matching the spectrum obtained with the parcel model. Due to application of this initialisation technique, the simulations are termed "parcel-DNS" approach.

The simulations are run with 12 different model settings that differ by enabling/disabling coalescence, turbulent fluctuations, solute effects and altering properties of the seeded droplets.

The problems addressed in the paper clearly match the scope of ACP. I concur with the first reviewer that Fig. 7 is a major riddle for the reader. Clearly, the piecewise-linear LWC profile needs to be explained and the "jumps in the statistics" need to be eliminated by deriving spectral properties from the droplet population and not from the binned spectrum.

I list below several other relatively major remarks that warrant requesting a major revision to the simulation protocol, result analysis and the manuscript itself.

1. First of all, I would argue that among all possible choices of the moment to switch on representation of turbulent inhomogeneities (i.e., the switch from parcel to DNS model), the level of peak supersaturation is the most unintuitive one. Numerically, it is likely one of the trickiest points for drop growth solver. Since the solute and curvature effects are resolved in the DNS, why not to benefit and resolve activation, especially as its sensitivity to supersaturation fluctuations is continuously being discussed in literature. It is all the more puzzling as the no-fluctuation activation is coupled with further growth in strongly turbulent environment.

[Figure]

2. The courageous assumption of 1.5 km adiabatic ascent with constant speed calls at least for more discussion on limitations of the study due to lack of representation of entrainment.

3. The numerical experiments presented in the paper lack any sensitivity analysis that would confirm the convergence of the results and quantify their sensitivity to spatial, spectral and temporal resolution as well as to the choice of set-up parameters. For instance, the initial aerosol spectrum is discretised onto a grid of only 39 classes for the parcel simulations, which is a crude resolution. While the Lagrangian-in-radius treatment of particle size evolution is indeed free from numerical diffusion (not dispersion - p5/l117), it is highly sensitive to the spectral discretisation (see e.g. discussion of Fig. 8 in Kreidenweis et al. 2003, doi:10.1029/2002JD002697).

4. Since the simulations feature collisional growth, perhaps it would be beneficial to analyse cloud and drizzle water separately (or is it already the case which could be related to the kink in the LWC profile in Fig. 7?), especially as the authors comment on autoconversion parameterisations. On a related note, the recent work by Noh et al. (2018, doi:10.1175/JAS-D-18-0080.1) is perhaps worth citing when discussing autoconversion rate dependence on spectral parameters (e.g., p8/l198,l204).

5. It would be beneficial to switch from reporting particle concentrations per unit volume to concentrations per unit mass of air, so the variation stemming from diminishing density along the 1.5 km ascent would be excluded. This could also help to understand the difference between the total particle concentration in the log-normal distributions $133 + 66.6 + 3.06 = 202.66\,cm^{-3}$ (in standard T,p conditions?) vs. total initial concentration of $112\,cm^{-3}$ (page 5, lines 112-113).

6. Mentioning seeding in the title of the paper would certainly better convey the focus of the study and, in my opinion, "an in-cloud seeding case study" could well

replace the "parcel-DNS approach" subtitle.

7. A table summarising the simulations would be very helpful. Currently, model description is mixed with the set-up description, while some key parameters are hard to find in the text (e.g., domain size is just given in parenthesis in a sentence on particle concentrations). Also, Table 1 would be more helpful with added "collisions" column and with all 12 simulations listed. Same concerns all mentions of "six experiments" - there are 12 DNS runs.

8. Last but not least, please clarify if the study can be independently reproduced by providing information on the versions of the model code used and its availability.

Other remarks:

- p1/l17: "interaction" ⇝ interactions

- p1/l34: space before parenthesis missing

- p2/l23: framework ⇝ frameworks

- p2/l26: "certain microphysical processes" – please be more specific

- p2/l38: "solve" ⇝ "solves"

- p2/l41: isn't the motivation to reduce the computational cost, rather than to reduce uncertainty? (replacing DNS with a parcel model actually increases uncertainty...), I would suggest removing the whole paragraph actually (lines 41–51)

- p2/l45: "aerosol processing" in some contexts is used to refer to modification of ambient aerosol after evaporation of droplets (due to aqueous chemical reactions and collisions) – perhaps worth rephrasing

- p3/l60: "nuclei ... enhances" ⤳ "nuclei ... enhance" (or "representation of ...")

- p3/l69: "Section 2.1-2.2" ⤳ "Sections 2.1-2.2"

- p3/l73: "droplet chemistry composition" ⤳ "hygroscopicity"

- p4/Fig2: suggest finding alternative wording for "stairs", please rephrase the last sentence: "fitting the distributions to the DNS" seems awkward, typo in "processers"

- p5/l109: are four significant digits really necessary when specifying initial RH?

- p5/l118: "thermodynamic equilibrium" sounds puzzling, I suggest following Jensen and Nugent and explaining what is meant: "in equilibrium (dr/dt=0)"

- p6/l136: "aerosol processing" – see comment p2/l45 above

- p6/eq2: drop growth equation (2) implies that supersaturation is defined as $S = e/e_s$ (as in Jensen and Nugent 2007), but in Chen et al. 2018b it is defined as $S = qv/q_{vs}$ – of course numerically almost the same, but perhaps worth clarifying

Added the definition when we introduced S below equation (1).

- p7/l167: why not replacing the inline fraction with just $\kappa = 0$?

- p7/l171: k ⤳ $\kappa$

- p7/l166: "turbulent advection of the supersaturation fluctuation" suggests $S'$ is among the advected quantities

It is correct that S is an advected quantity since S is determined by two passive scalars: T and Qv, both are advected by turbulence

- p8/184: "extremely slow": be more specific

We have modified the paragraph based on the new simulated results and thus removed this sentence.

- p8/187: "when" ⤳ "When"

- p8/l192: $o()$ ⤳ $\mathcal{O}()$
- p8/l197: space before parenthesis

- p8/l202: avoid word "claim"

- p9/Fig2: mention in the caption that collisions were enabled

- p11/Figs5-6: mention in the caption that collisions were enabled

- p12/Fig7: mention in the caption that collisions were enabled

- p14/l297-298: remove "which is a major facility"?

- p14/l300-301: rephrase "support from Cheyenne ... and from Graham and Cedar"

- References: use journal abbreviations

- References: most entries have doi/url given twice

- References: if there is a doi assigned, do not list url (e.g.: Skamarock et al., Yang et al.)

---

## Author Comment (AC1) · 11 May 2020

We greatly appreciate the invaluable comments from the reviewers. Below are the point-by-point responses. The comments are shown in black and responses are in blue.

**Response letter to reviewer #1**

**Major Comments**

**Previous literature**

The broadening of droplet size distributions has been studied intensively in the past, including the effects of aerosol hygroscopicity and turbulence (supersaturation fluctuations). Of course, the representation of these processes was highly idealized in the past, but the conclusions are not different from the submitted manuscript. For instance, Srivastava (1991) showed that the consideration of curvature in the diffusional growth equation is essential for the spectral broadening of a droplet size distribution in a lifted parcel. Furthermore, Korolev (1995) showed that curvature and solute terms lead to irreversible broadening if supersaturation fluctuations are applied. Further analysis has been carried out by Çelik and Marwitz (1999). Integrating these very simple, even analytical investigations into the highly complex framework of the submitted study will increase physical understanding and insight.

We integrated the above literature into our manuscript in Section 2.2 (Line 153-160): "Parcel model studies on droplet condensation in a lifted parcel show that the curvature term and the solute term can lead to condensational broadening on the droplet size spectrum. Srivastava (1991) demonstrated that the curvature effect is essential for DSD broadening in an ascending parcel. Korolev (1995) found that the curvature effect and the solute effect lead to irreversible broadening when supersaturation fluctuations are present. It is also found that aerosols of different sizes and different hygroscopicity can cause spectral broadening without supersaturation fluctuations (Çelik and Marwitz, 1999; Jensen and Nugent, 2017). Therefore, it is crucial to examine whether these effects are important in spectral broadening when they dynamically couple with droplet collisional growth in a turbulent environment. "

**Analysis**

Many conclusions are based on Fig. 7. While the authors are candid about the fact the mean radius (panel a), the dispersion (panel b), the maximum radius (panel c), and the radar reflectivity (panel e) are calculated from droplet size distribution, I believe that this is also the case for the liquid water content (LWC) in panel d. Since the parcel is ascending with a constant velocity, the LWC is expected to increase linearly with time:  $LWC = w t \Gamma$ , with  $\Gamma \approx 2.0 \times 10^{-6}$  kg m^-4. While the LWC )) follows this linear relationship for the first three minutes, it deviates significantly afterward. Of course, the LWC may deviate from this adiabatic behavior if the transfer of water vapor into the liquid phase is kinetically limited. However, this effect can be excluded since its magnitude decreases with droplet size and hence time. One reasonable explanation for this unphysical behavior is that the authors diagnosed the LWC from the droplet

size distribution (as also suggested by the similarity to the mean radius in panel a) or made another error in the calculation of the LWC. Accordingly, any conclusions on LWC differences between the individual model runs are potentially false (e.g., II. 9 - 10, 231 - 240, 265 - 268). Therefore, I strongly suggest repeating all simulations of this study, and to diagnose the LWC directly from the simulated droplets. I further suggest doing the same for the mean radius, dispersion, and maximum radius.

We appreciate the reviewer for pointing out this defect in our study. We identified that the unphysical behavior of the LWC and mean radius was caused by the round-off & truncation error of the single-precision droplet radius when solving the droplet growth equation. We have fixed the issue and re-run all 12 simulations in the revision.

The droplet growth equation in the present study includes the curvature effect and the solute effect. In previous version in Chen et al., (2018b, eq. (B1)), the equation we solved was  $\frac{dR^2}{dt} = 2K f_v S$ , where  $K^{-1}$  is a temperature-dependent coefficient. In the present study, the  $\frac{dR^3}{dt} = 2K f_v S$

droplet growth equation is modified to  $\frac{dR^3}{dt} = 3Kf_v(S - f(solu, curv))$ , where f(solu, curv) $dR^3$

includes the curvature term and the solute term. The dt format was used in the model for the convenience of calculating the condensation rate. It is more sensitive to the precision of

 $dR^2$

calculation than  $\overline{dt}$  because it's in a higher order. It should be pointed out that this truncation error issue only happened in the present study and did not affect the previous studies when the equation  $\frac{dR^2}{dt} = 2Kf_vS$  was used. As shown in Fig. A, when using the  $\frac{dR^2}{dt}$  scheme, the LWC

equation  $dt = 2\pi f_v dt$  was used. As shown in Fig. A, when using the dt scheme, the LWC evolution is not truncated by a lower precision.

**$dR^3$**

In the dt scheme when supersaturation reaches below a certain critical value, a non-negligible round-off error occurred in calculating the new  $R^3$ :  $R^3_{new} = R^3_{old} + dR^3$ .  $dR^3$  is rounded off to 0 when it is added to  $R^3_{old}$ , causing  $R^3$  to stop growing (see dashed lines in Fig. B below). This caused the LWC and other spectra-derived statistics to plateau around T= 2min.

We switched all droplet-related variables to double-precision to minimize the round-off error and have rerun all simulations accordingly. In the new version,  $R^3$  grows linearly with time (solid lines in Fig. B). All contents in the revised manuscript are based on the results of the new simulations.

Fig. A: the time evolution of LWC when using the droplet growth equation  $\frac{dR^2}{dt} = 2K f_v S$  (no solute effect or curvature effect). The green line is based on double-precision calculation, and the red line is based on single-precision calculation.

Fig. B: the time evolution of the mean  $R^3$  and  $dR^3$  in Run A.  $dR^3$  is the difference of  $R^3$  from the previous time step. Dotted lines are from the single-precision run, and solid lines are from the double-precision (new version) run.

**Limitations of the Modeling framework**

The authors state that the applied modeling framework promotes a more realistic assessment of cloud microphysical processes during the early development of a cloud. And I agree. However, there are certain limitations and restrictions inherent to this approach that need to be addressed. In fact, they might restrict the scope of the manuscript's conclusions significantly. **First**, the ascent of the air parcel is adiabatic. In cumulus clouds, entrainment dilutes the cloud constantly, reducing the liquid water content, while the activation of newly entrained aerosols might lead to a further broadening of the droplet size distribution (e.g., Lasher-Trapp et al. 2005).

We thank the reviewer's comments and have addressed the limitations and restrictions of this approach in the revised manuscript.

We agree that entrainment is a key process in cumulus clouds, in particular, close to cloud edges. Our study mainly looks at the adiabatic core region where entrainment mixing is minimum. It is found in Khain et al. (2013) that this region contains more large droplets due to higher liquid water content (LWC) than the rest of the cloud and is argued to favor the formation of raindrops.

In the revision, we added the description of the adiabatic region and explained the importance of this region to rain initiation in the introduction (lines 67-71):

"The adiabatic cores are regions free of entrainment of dry air. This region has a higher liquid water content (LWC) than the rest of the cloud and is argued to favor the formation of raindrops (Khain et al., 2013). To represent the DSD evolution at the core region, we prescribe here a dry aerosol size distribution in the sub-cloud region, and the aerosol activation process is explicitly simulated by a parcel model to provide a more physically-based initial DSD for the DNS."

In the result section, we discussed the supersaturation fluctuations in adiabatic core regions and the impact of the entrainment on condensational broadening at cloud edge on lines 219-224:

"Even though turbulence intensifies the collisional growth, the modulation on the droplet condensation is found insignificant. The DSDs in Run A and B in the condensation-only set are nearly identical (Fig. 4 (d-e)). This is because the supersaturation fluctuations are weak in an adiabatic core region. Vaillancourt et al. (2002) found that in a quasi-adiabatic environment both particle sedimentation and short-lived turbulent coherent structure reduce the supersaturation fluctuations and decrease the time that droplets are exposed to these fluctuations. We expect that the turbulent-induced condensational broadening is more significant in the cloud edge where entrainment mixing induces large variation in supersaturation fluctuations."

**And lines 254-255:**

"Thirdly, our idealized simulation focuses on the cloud adiabatic core which is devoid of entrainment. Inhomogeneous mixing by entrainment can possibly broaden the DSD."

In the conclusion section, we addressed the limitations inherent to this approach and the importance of including entrainment in future investigations on lines 339-350:

"Our idealized simulations focus on the cloud adiabatic core region and therefore exclude entrainment mixing which is highly active near the cloud edge. The activation of newly entrained aerosols might lead to a further broadening of the DSD (LasherTrapp, et al., 2005). In addition, the in-cloud mixing at a much larger scale than the DNS domain transports and mixes both the air and droplets from different parts of the cloud including the cloud edge, leading to a highly perturbed Lagrangian supersaturation experienced by droplets (Grabowski and Abade, 2017, eddy hopping effect). On the other hand, larger turbulent eddies can generate higher supersaturation fluctuations due to a higher variation in a vertical motion and thus may both affect the aerosol activation and broaden the DSD. Traditional DNS which is confined to a relatively small domain size (<1 m), and the impact of supersaturation fluctuations is significantly restricted. Methods such as an up-scaled DNS with superdroplets (e.g., Thomas et al., 2020) or representing the large-scale mixing with an external forcing on the thermodynamic fields (Paoli and Shariff, 2009) can be used for studying the impact of turbulent scales on the supersaturation fluctuations and thus on the condensational broadening of DSD. In conclusion, the relative importance of entrainment, eddy hopping effect, small-scale turbulence and aerosols requires further investigation."

**Second**, the kinetic energy dissipation rate is assumed to be constant throughout the ascent, using a relatively high value only typical for the top of shallow cumulus clouds. This is a crude simplification since turbulence tends to increase with height (e.g., Seifert et al. 2010).

We used a non-turbulent parcel model to handle the ascending process below the cloud base, and use a statistically-stationary dissipation rate in DNS above the cloud base. The advantage of this idealized, simplified treatment is that the effect of turbulence can be easily distinguished and quantified from other effects/processes. We have addressed this limitation in the manuscript (lines 186-190): "It should be pointed out that the dissipation rate in cumulus clouds tends to increase with height. For simplicity, the eddy dissipation rate ( $\epsilon$ ) for all the turbulent cases is set statistically stationary ( $\epsilon = 500 \ cm^2 \ s^{-3}$ ). The advantage of this idealized, simplified treatment is that the effect of turbulence can be easily separated from aerosol effects. A dissipation rate of  $500 \ cm^2 \ s^{-3}$  represents a strongly turbulent environment in cumulus clouds to examine the upper-bound of turbulent effects on the DSD evolution."

**Fourth,** the DNS domain is relatively small. Restricting vertical motions to 16.5 cm, the impact of supersaturation fluctuations is significantly restricted (e.g., Abade et al. 2018).**

On the one hand, the turbulence in the adiabatic region is nearly homogeneous and isotropic (Vaillancourt et al. 2002), and the supersaturation fluctuations at local scales are mainly induced by droplet condensation and evaporation. We added a discussion on the impact of small-scale turbulence on supersaturation fluctuation (lines 220-224):

"... the supersaturation fluctuations are weak in an adiabatic core region. Vaillancourt et al. (2002) found that in a quasi-adiabatic environment both particle sedimentation and short-lived turbulent coherent structure reduce the supersaturation fluctuation and decrease the time that droplets are exposed to these fluctuations. We expect that the turbulent-induced condensational broadening is more significant in the cloud edge where entrainment mixing induces large variation in supersaturation fluctuations."

On the other hand, the fluctuations in the vertical motions of scales larger than traditional DNS may be important to perturb the supersaturation. We discussed the limitation of DNS in the revision on lines 341-350:

"In addition, the in-cloud mixing at a much larger scale than the DNS domain transports and mixes both the air and droplets from different parts of the cloud including the cloud edge, leading to a highly perturbed Lagrangian supersaturation experienced by droplets (Grabowski and Abade, 2017, "eddy hopping effect"). On the other hand, larger turbulent eddies can generate higher supersaturation fluctuations due to a higher variation in a vertical motion and thus may both affect the aerosol activation and broaden the DSD. Traditional DNS which is confined to a relatively small domain size (<1m), and the impact of supersaturation fluctuations is significantly restricted. Methods such as an up-scaled DNS with superdroplets (e.g., Thomas et al., 2020) or representing the large-scale mixing with an external forcing on the thermodynamic fields (Paoli and Shariff, 2009) can be used for studying the impact of turbulent scales on the supersaturation fluctuations and thus on the condensational broadening of DSD. In conclusion, the relative importance of entrainment, eddy hopping effect, small-scale turbulence and aerosols requires further investigation."

**Fifth**, the range of investigated aerosol concentrations is relatively narrow. Especially the investigation of even lower aerosol concentrations might be of interest to assess the relative importance of aerosol hygroscopicity and turbulence even further.

We have included the discussion of the range of aerosol concentration in the revision:

"However, the seeding particles in this study only cover a limited range of dry radius (  $R = 0.1, 1 \ \mu m$ ) and number concentration ( $N = 10, \ 20 \ cm^{-3}$ , corresponding to 10 - 20%increase in the total number concentration). Conditions with more ultra-giant aerosols (  $R \gg 1 \ \mu m$ ), lower aerosol concentrations ( $N \ll 100 \ cm^{-3}$ ), or highly polluted environment (  $N \gg 100 \ cm^{-3}$ ) will be of interest to further assess the relative importance of aerosols and turbulence."

**Minor Comments**

LI. 3, 69: The term "aerosol loading" feels ambiguous here. I believe you want to investigate the effects resulting from changes in the number of CCN.

**We have changed the aerosol loading to aerosol number concentration for clarification.**

L. 8: Inhibiting is a very strong word here. As long as the smaller droplets are activated and the domain is supersaturated, their growth might be reduced but not inhibited.

We replaced "inhibiting" with "reducing".

**L. 9: "[...] seeding reduces the LWC [...]". See major comment and revise accordingly.**

We have addressed the unphysical behavior of LWC and rerun all the simulations. The new results show that LWC grows linearly with height. It is also found that the LWC is insensitive to aerosols and turbulence (Fig. 7 (a)). We have rewritten the discussion and conclusion based on the new results. For instance, lines 256-277 discussed the insensitivity of LWC to turbulence and aerosols and the implication on autoconversion parameterizations.

LI. 9 ff.: The term *effective radius* is used interchangeably with *mean radius* within the entire manuscript. While the *mean volume radius* is usually close to the effective radius, this is not necessarily the case for the *arithmetic mean radius*. Please clarify.

We replaced all *effective radius* with *mean radius*. In the revision, we refer *mean radius* only to *the arithmetic mean radius* throughout the manuscript for consistency.

Ll. 35 – 36: The comment regarding the effects of cloud chamber walls is appreciated. However, I feel that this relatively specific effect might demand a reference, e.g., Thomas et al. (2019).

We expanded the argument to include more detail in wall effects on lines 33-37:

"On the other hand, laboratory facilities such as cloud chambers are difficult to create environments scalable to real clouds. Furthermore, the effects of chamber walls, such as the heat and moisture fluxes fed into the solid wall and the droplet loss due to their contact with the wall, are challenging to quantify with considerable uncertainties in the measurements. For example, Thomas et al. (2019) used a flux-balance model to estimate the wall effect on the mean temperature and mean water vapor mixing ratio and found that the results highly depend on the geometry of the chamber."

LI. 43 – 44, 89: Why is it necessary to divide the model into the two steps using a standard parcel model first and subsequently a DNS? I assume it is straightforward to consider the activation of aerosols within the DNS framework, similar to the parcel model. In fact, considering turbulent supersaturation fluctuations during activation might have important implications for the estimated number of cloud droplets (Abade et al. 2018). Overall, this could result in a more consistent modeling approach. And based on the data provided in the manuscript, one could easily estimate that the total computing increases by less than 30 % when the DNS is used for the entire ascent. I consider this increase as acceptable.

We agree that using a DNS model for the entire process will yield a more consistent and more accurate result. One of the purposes of this paper is to reduce the computational load, as not all aerosols are activated, and the unactivated aerosols have little impact on the supersaturations or the DSD. This approach is particularly useful and economical to deal with the situation of a large number of aerosols in the beginning and a small number of activated droplets subsequently.

In addition, it is discussed in the paper that the supersaturation fluctuations at DNS scales are small and therefore have a limited impact on the droplet activation and droplet condensational growth. The eddy-hopping effect discussed in Abade et al. (2018) mainly comes from large scale turbulence due to the fact that the variance of supersaturation increases with the turbulence scales. For the scales of DNS (less than 1m), the fluctuation/variance of supersaturation is too small to see the effect. This argument is also supported by Vaillancourt et al. (2002) which found that small-scale turbulence has a negligible effect on droplet condensational growth. The above discussion is included in the conclusion section (lines 341-350).

**L. 84: Add the height of the maximum supersaturation to the text.**

Included on lines 84-86:

"The first phase starts from the unsaturated sub-cloud region ( $\approx 300 \ m$  below cloud base) to the level where the supersaturation reaches a maximum ( $\approx 43 \ m$  above cloud base, see Fig. 2(a))."

Caption to Fig. 2: Is it really the relative altitude? It is still the (absolute) height above cloud base. A *relative* altitude is usually divided by another quantity for normalization.

To avoid confusion, we switched to "the height from cloud base  $(H - H_{CB})$ "

LI. 92 – 93: The fact that droplets smaller than 10  $\mu$ m in radius collide rarely is a well-known fact, and can be obtained from any text book on cloud physics.

This argument is true in the context of a still-air case but needs to be carefully interpreted for turbulent cases. We modified the sentence to "These droplets have very small collision rates even in strong turbulence" for clarification.

L. 94: I believe that the hygroscopicity parameter is represented by  $\kappa$  in the Eq. (1) and (2). Am I right? I suggest stating this explicitly.

Yes, we added  $\kappa$  on line 101, just before introducing equation (1).

Ll. 123 – 124: How do you distinguish between activated and unactivated aerosols? Large aerosols do not need to be activated (i.e., be larger than the critical radius) to behave like a droplet.

We first identified the smallest activated aerosol size by comparing it with its critical radius. All aerosols equal to or larger than this size are treated as "activated" and all aerosols below this size are "unactivated". In this way, giant aerosols, even though they may not be activated are included in the DNS.

**LI. 126 – 129: Does this manipulation also change the liquid water content in the DNS simulation?**

The change of liquid water content due to the fitted size distribution is negligible as the total number of droplets from each bin of the parcel model is much larger than the number of processors (=64) used in the DNS. The resulting difference in the LWC is 0.002g/kg at the initial time of DNS simulation (0.028 g/kg in the DNS and 0.030g/kg in the parcel model).

Ll. 136 – 139, Eq. (2): Why don't you use the same diffusional growth equation for the parcel and the DNS model?

The equation (2) is almost the same as (1) except that the DNS version considers 1) the ventilation effect (reflected in fv coefficient) and 2) uses instantaneous S and T at droplet location instead of the parcel mean. In the revision, we removed equation (2) and discussed the differences between the parcel model and the DNS model when applying the equation in Section 2.1 (lines 110-115).

Table 1: I suggest using simpler names for the model runs, indicating directly the difference to the control run. E.g., "run NoTurb" instead of "run B".

We appended simple straightforward names (CTL, NoTurb, NoSolu, Seed\_1N1R, Seed\_1N2R, and Seed\_2N1R) in Table 2 as well as in the context to explicitly indicate the difference between the six cases.

LI. 165 – 166: This is misleading since the particles are still lifted with a mean updraft of 2 m s-1.

We have modified the description to "When turbulence is switched off, the background velocity fluctuation is set to  $0 m s^{-1}$ . Therefore, particle motion is only affected by the mean updraft and gravitational settling..."

L. 166: What is meant by "turbulent advection of the supersaturation fluctuation"? Is there no turbulent mixing in the DNS domain, only molecular diffusion, when "turbulence is switched off"? This might overestimate the effects of supersaturation fluctuations caused by processes other than turbulence, e.g., the faster depletion of supersaturation in the vicinity of a large droplet.

Yes, when turbulence is off, there is no turbulent "mixing" of the supersaturation field, though advection will be a more precise term, as turbulence does not truly mix the field which enhances the subsequent mixing by diffusion. As such, the local supersaturation fluctuation is only affected by molecular diffusion and droplet condensation/evaporation.

To reduce the confusion, we changed the statement to "Therefore, particle motion is only affected by the mean updraft and gravitational settling, and the supersaturation fluctuation is only induced by droplet condensation and evaporation." (lines 178-180)

L. 174: A kinetic energy dissipation rate of 500 cm2 s-3 at cloud base is too high. Typically, the dissipation rate increases with distance to the cloud base, and a value of 500 cm2 s-3 might only be representative for the top of a shallow cumulus cloud.

We addressed this question above under the second comment of the *Limitations of the Modeling framework*.

Sec. 3: I suggest introducing subsections to increase clarity of this section.

We divided the result section into two subsections: 3.1 natural cases and 3.2 seeded cases.

L. 177: You state that the parcel rises up to 1.2 km above cloud base. This ascent lasts 600 s = 10 min, assuming an updraft of 2 m s-1. Accordingly, the analysis after 6 min does not constitute the end of the simulation. Moreover, it is also not clear if this point in time is considered to be after the start of the DNS or after the start of the parcel model.

We thank the reviewer for pointing this out. The calculation was wrong and we corrected it in the revision (line 89)

In the revision, we have extended the simulated time to 500s (Table 1) which gives us a lifting height of 1 km.

L. 184: "Condensational growth after 1 min becomes extremely slow [...]". "How can we distinguish between condensational growth and collision-coalescence in Fig. 5b (Fig. 3b in the revision)?"

We removed the statement, and the new discussion on distinguishing the condensational growth and collisional growth is based on the relative dispersion in Fig. 7 (c) (lines 243-245):

"Condensational growth narrows the DSD and decreases the relative dispersion in the condensation-only set (dotted lines in Fig. 7(c)). Droplet growth in the first two minutes is prevailed by condensation, as the relative dispersion in the condensation-collision set of experiments well overlaps with that in the condensation-only set."

L. 187, Fig. 7b (Fig. 7(c) in the revision): The displayed dispersion values are much smaller than the values observed in real clouds. In fact, they are about one order in magnitude smaller. You should comment on this difference and name reasons.

In the previous version, the value of relative dispersion in some cases is << 0.1. The low values are mainly caused by the single-precision issue that truncated the growth of the large droplets (see Fig. A and the discussion under the comment of *"Analysis"*). This round-off error issue led to an overestimation of condensational narrowing and thus a lower dispersion. The issue also caused the LWC, mean radius, etc reached a plateau after a certain time point, which is unphysical. To solve this issue, we have switched all droplet-related variables to double precision.

In the new simulations, the dispersion in the condensation-collision set of experiments is around 0.1 at the end of 500s and is expected to grow.

In the revision, we listed a few reasons to explain the lower value in dispersion than observed in real clouds (lines 251-255):

"It is recognized that the relative dispersion of around 0.1 in this study is smaller than observed in most flight measurements. Firstly, the flight measurement is an average of a long-distance sampling ( $\mathcal{O}(100m)$ ) which does not capture the local property of droplet size distribution and therefore is not comparable to our modeled results. Secondly, the simulations only last for 500s, and it is expected that the relative dispersion keeps growing. Thirdly, our idealized simulation is focused on the cloud adiabatic core which lacks entrainment. Inhomogeneous mixing by entrainment can possibly broaden the DSD."

Ll.188 – 190: What is meant by "multi-modal feature"? For me, the droplet spectra seem to be almost monomodal. There might be a second mode developing between 20 and 25  $\mu$ m, but it is not very distinct.

We agreed that the multi-modal feature is not significant. Therefore, we removed the description.

Fig. 3e : After 6 min, small (radius < 1  $\mu$ m) droplets seem to appear in the droplet size distribution. Where do they come from?

We have rerun all simulations to correct the precision issue, and the new results did not show any small radius  $< 1 \mu m$ .

LI. 210 ff.: By CCN case you mean the case with solute effects (but no turbulence) or the control case?

It refers to the control case with both CCN and turbulence. To clarify, we have changed the description to "Run A (CTL)", "Run B (NoTurb)", and "Run C (NoSolu)" to refer to the control case, non-turbulent case, and no solute case, respectively and updated Table 1 accordingly.

**LI. 221 – 223: In what process are the hygroscopic CCN more effective in the first few minutes?**

The original statement was not very clear. We removed the statement and modified the discussion on the effect of CCN hygroscopicity on lines 291-299:

"Finally, aerosol hygroscopicity is key to the onset time of autoconversion. All five aerosol-embedded cases see a similar onset time around T= 4 min. Removing the solute effect (hygroscopic material) in Run C delays the onset of autoconversion by about 1.5 min (green line Fig. 7 (f)). Nevertheless, after T = 6-7 min, the autoconversion rate in Run C exceeds all seeded cases. First, solute (CCN hygroscopicity) has a negligible effect on the growth of small aerosols, as the size distribution of small droplets in Run A and C remain almost identical. This is substantiated by the almost identical collision frequency of droplets below  $20 \ \mu m$  of the two cases (Fig. 6 (a-c)). Second, seeding reduces the mean radius of the droplets. This leads to a reduction in collisions for droplets over  $20 \ \mu m$  (Fig. 6(d)) and subsequently decelerates the autoconversion process. The above findings imply that increasing the aerosol size (ultra-giant aerosol) shortens the lifetime of the clouds through a fast onset of rain. And increasing the number of aerosols decelerates the rain process."

**L. 226: Define large droplets by their size range.**

**We removed the original discussion in the revision.**

LI. 234 – 238: It is impossible that a higher droplet concentration (as a result of the seeding) results in a lower liquid water content. In fact, in a rising parcel without (interactive) entrainment, one would assume a higher liquid water content due to the accelerated depletion of water vapor. See major comments above.

In the new simulations, a similar LWC is observed in both the unseeded and seeded cases. The value is only marginally higher in the seeded cases, but the difference is negligible.

L. 243: A "flatter and broader" droplet size distribution sounds tautological. Since you do not change the number of CCN significantly, a broader distribution needs to be flatter. Or does "flat" refer to another property of the distribution that I miss?

**We have removed "flatter" and only used "broader".**

LI. 231 – 247: All seeding experiments tend to address the effect of additional CCN. Those will always decrease the condensational growth of individual droplets since the water is distributed

on a larger number of droplets. The more interesting case would be a reduction in the number of CCN.

A reduction in the number of CCN is arguably the opposite of adding additional CCN. In that sense, we can use the CTL case as a reduced-number case.

Fig. 7: The colors are barely distinguishable. Please change them. Why do you show the radar reflectivity (panel e), which is not discussed in the text?

We have changed the color scales and removed the radar reflectivity from the panel in Fig. 7 and added the autoconversion rate (f) and maximum radius (b).

LI. 265 – 268: This conclusion is based on inadequate analysis and is not true as outlined in the major comments.

We have modified the conclusion based on the new results and updated discussion of LWC and autoconversion on lines 256-273.

LI. 270 – 272: This claim is only true because of the limited range of CCN concentrations tested in this study. For the analyzed cloud, it will probably always rain if the CCN concentration is reduced to 10 cm-3 and it will probably never rain if it is increased to 1000 cm-3, irrespective of turbulence or aerosol hygroscopicity.

We agreed that the argument of this study holds under the condition of the limited range of CCN we tested. And we addressed this limitation on lines 315-319:

"However, the seeding particles in this study only cover a limited range of dry radius (  $R = 0.1, 1 \ \mu m$ ) and number concentration ( $N = 10, \ 20 \ cm^{-3}$ , corresponding to 10 - 20%increase in the total number concentration). Conditions with more ultra-giant aerosols ( $R \gg 1 \ \mu m$ ), lower aerosol concentrations ( $N \ll 100 \ cm^{-3}$ ), or highly polluted environment ( $N \gg 100 \ cm^{-3}$ ) will be of interest to further assess the relative importance of aerosols and turbulence."

**Technical Comments**

Language: While the language is understandable, several smaller mistakes slow down the reading process.

L. 3: Define DNS in the abstract.

Defined

**L. 39: Remove "Lagrangian-tracking".**

**Removed**

L. 43: DNS has already been defined in I. 36.

**Removed the definition**

L. 56: It is odd citing Chen et al. (2018b) before citing Chen et al. (2018a).

We arrange the order according to the publication time in which 2018a was published before 2018b.

L. 61: I suggest adding "subsaturated" before downdrafts.

**added**

L. 81 ff.: Units are usually stated in upright characters and separated by a thin space (\, in LaTeX) from the numerical value.

**corrected**

L. 107, several other occasions: I suggest using exponents for units (m s-1 instead of m/s), consistent with the notation used throughout the manuscript.

We modified the units for consistency.

L. 200: Why do you introduce the abbreviation BR74, which is used only once in the manuscript?

Fig. 7d: It is g/kg, and not g/Kg.

We changed it to g/kg.

**Response letter to reviewer #2**

The problems addressed in the paper clearly match the scope of ACP. I concur with the first reviewer that Fig. 7 is a major riddle for the reader. Clearly, the piecewise- linear LWC profile needs to be explained and the "jumps in the statistics" need to be eliminated by deriving spectral properties from the droplet population and not from the binned spectrum.

We have identified the error causing the unphysical behavior of the LWC profile. See detailed discussion in the response to reviewer 1 (see *Analysis* under Major Comments). In the new simulations, the LWC follows a linear trend.

We have replotted Fig. 7 based on the calculation from the droplet population instead of from the binned spectrum.

I list below several other relatively major remarks that warrant requesting a major revision to the simulation protocol, result analysis and the manuscript itself.

1. First of all, I would argue that among all possible choices of the moment to switch on representation of turbulent inhomogeneities (i.e., the switch from parcel to DNS model), the level of peak supersaturation is the most unintuitive one. Numerically, it is likely one of the trickiest points for drop growth solver. Since the solute and curvature effects are resolved in the DNS, why not to benefit and resolve activation, especially as its sensitivity to supersaturation fluctuations is continuously being discussed in literature. It is all the more puzzling as the no-fluctuation activation is coupled with further growth in strongly turbulent environment.

We agree that performing a full DNS experiment and including the activation stage is most beneficial. The main focus of this paper is on the subsequent droplet growth after the activation stage. And the parcel-DNS framework provides an economical approach when the sub-cloud aerosol number concentration is much higher than the cloud droplet number concentration. By filtering out the unactivated aerosols, DNS can largely reduce the computation in tracing individual particles.

We justified this treatment on lines 92-97:

"Outputs from the parcel model at the height with maximum supersaturation are fed into DNS as initial conditions. Because unactivated aerosols have little influence on the subsequent droplet growth or on the water vapor field, only the activated aerosols from the parcel model are carried over to the DNS model as the initial background aerosol condition to decrease the computational load. The CCN size distribution and droplet size distribution are displayed in Fig. 2(c). This parcel-DNS hybrid model provides an economical approach and is the first step towards a fully DNS-resolved simulation of the entire ascending process."

In the conclusion section (lines 351-355), we bring up the importance of implementing a full DNS from below the cloud base to include the effect of turbulence (and supersaturation fluctuations) on aerosol activation.

"This study proposes the first DNS model framework for scrutinizing the microphysical impact of cloud seeding and presents the first results of such a model. Full DNS modeling from below the cloud base will be the next step to include the effect of turbulence on aerosol activation. Additionally, more realistic scenarios resembling actual hygroscopic seeding conditions, such as utilizing multi-disperse size distributions, different hygroscopicity parameters, and seeding below the cloud base will be designed in the future development and deployment of this framework."

2. The courageous assumption of 1.5 km adiabatic ascent with constant speed calls at least for more discussion on limitations of the study due to lack of representation of entrainment.

In the revision, we added more discussion about the limitation of a lack of entrainment in our model throughout the article, and in the meantime stressed that our study focuses on the adiabatic core region.

Please see detailed discussion in the response to reviewer 1 under the first point of "*Limitations* of the Modeling framework"

- 3. The numerical experiments presented in the paper lack any sensitivity analysis that would confirm the convergence of the results and quantify their sensitivity to spatial, spectral and temporal resolution as well as to the choice of setup parameters. For instance, the initial aerosol spectrum is discretised onto a grid of only 39 classes for the parcel simulations, which is a crude resolution. While the Lagrangian-in-radius treatment of particle size evolution is indeed free from numerical diffusion (not dispersion p5/l117), it is highly sensitive to the spectral discretisation (see e.g. discussion of Fig. 8 in Kreidenweis et al. 2003, doi:10.1029/2002JD002697).
- 1) Sensitivity on the spatial and temporal resolution has been tested in Chen et al. (2016) for studies without hydrodynamic interactions and in Chen et al (2018) for studies with hydrodynamic interactions:

The spatial resolution (dx) of the flow field is confined to be smaller than the Kolmogorov length scale (the smallest size of turbulent eddies), see detailed discussion in Chen et al. (2016, p625). In this way, all energy-containing eddies affecting the droplet motion are well-resolved.

The temporal resolution (dt) is chosen to satisfy the CFL condition of the flow and at the same time well below the droplet inertial response timescale for an accurate representation of droplet trajectory, i.e.,  $dt = min(dt_{drop}, dt_{flow})$ .  $dt_{flow}$  has to maintain courant number <0.25, and  $dt_{drop} = 0.15\tau_{pmin}$ , and  $\tau_{pmin}$  is the droplet response time of the smallest inertial droplet in the domain. See Chen et al. (2018, Section 2e) for detailed validation.

2) Sensitivity on the bin resolution:

The 39 classes applied in the parcel model follow the same bin resolution as Xue et al. (2010). The bins are discretized on a log scale with the bin width increased by doubling the mass., i.e.,  $r(n)^3 = r(1)^3 * 2^{n-1}$ . In this way, the resolution is higher at small

particle sizes and lower at large particle sizes. This configure is reasonable since the large size has a lower number concentration.

As increasing bin resolution reduces the number concentration for each bin, the bin width should be large enough that at the end we can assign a reasonably large integer number of particles of each size in the DNS domain. With our current resolution, for  $r > 3\mu m$ , each bin corresponds to less than one particle in each processor of the simulation (the number of droplets is evenly distributed across every processor initially); for  $r_{dry} > 1 \ \mu m$ , each bin corresponds to less than 8 particles; for  $r_{dry} > 0.4 \ \mu m$ , each bin has less than 100 particles.

At the small size end, we agree that the resolution will impact the activation rate of the aerosol, and thus the number of droplets present in the domain will vary. Fig. C below shows the number of concentration of unactivated aerosols varying with bin resolution. The number of bins spans from 32 to 253, corresponding to a multiplication factor of radius between two consecutive bins from 2.2 to 1.1. The red marker shows our current model configuration of 39 bins with a multiplication factor of 2. The total number of unactivated aerosols varies more in the range between 32 - 67 bins. However, 39-bin configuration produces a similar result with the higher resolution cases ( $N = 19.66 \ cm^{-3}$  in 39-bin vs  $N = 20.25 \ cm^{-3}$  in 253-bin).

*Fig. C: Sensitivity of the number concentration of unactivated aerosols to the number of bins using a multiplication factor from 2.2 to 1.1. The red marker shows our current model configuration of 39 bins with a multiplication factor of 2.*

In the end, we stress that the bin resolution in our simulation only matters when comparing to the real observation of aerosol size distribution. First, the continuous lognormal distribution is only the approximated representation of the aerosol size distribution in one maritime condition. Adding the resolution does not refer to a more accurate representation of the observation.

Second, the moving-bin scheme does not produce numerical diffusion of the size distribution (we replaced the word "dispersion" with "diffusion" in the manuscript). It follows that the resolution of the size spectrum will also have no numerical impact on the evolution of the droplet spectrum.

4. Since the simulations feature collisional growth, perhaps it would be beneficial to analyse cloud and drizzle water separately (or is it already the case which could be related to the kink in the LWC profile in Fig. 7?), especially as the authors comment on autoconversion parameterisations. On a related note, the recent work by Noh et al. (2018, doi:10.1175/JAS-D-18-0080.1) is perhaps worth citing when discussing autoconversion rate dependence on spectral parameters (e.g., p8/I198,I204).

The kink was due to the single-precision round-off error, and in the new simulations, the LWC stays linear with height.

We have added the autoconversion rate in Fig. 7. And cited Noh et al. (2018) in the discussion (line 264-265).

5. It would be beneficial to switch from reporting particle concentrations per unit volume to concentrations per unit mass of air, so the variation stemming from diminishing density along the 1.5 km ascent would be excluded. This could also help to understand the difference between the total particle concentration in the log-normal distributions 133 +  $66.6 + 3.06 = 202.66 \ cm^{-3}$  (in standard T,p conditions?) vs. total initial concentration of 112  $\ cm^{-3}$  (page 5, lines 112-113).

The number concentration of [133, 66.6, 3.06] mentioned in the manuscript are parameters used for defining the three log-normal distributions. And the aerosol size distribution used in the simulation only took a limited size range. N=202.66  $cm^{-3}$  is the total number concentration over the entire size range ( $r = 0 - \infty \mu m$ ). The aerosol size range we chose was from  $r = 10^{-3} - 49 \ \mu m$ , resulting in 112  $cm^{-3}$  in total concentration. The difference in concentration per unit volume due to diminishing density is small for an ascending of 1km (< 10%), therefore, we will still keep per unit volume as it is a preferably customary unit in both cloud measurement and modeling communities.

6. Mentioning seeding in the title of the paper would certainly better convey the focus of the study and, in my opinion, "an in-cloud seeding case study" could well replace the "parcel-DNS approach" subtitle.

We have changed the subtitle to "An in-cloud seeding case study using a parcel-DNS approach" to address both the physical process and the method.

7. A table summarising the simulations would be very helpful. Currently, model description is mixed with the set-up description, while some key parameters are hard to find in the text (e.g., domain size is just given in parenthesis in a sentence on particle

concentrations). Also, Table 1 (Table 2 in the revision) would be more helpful with added "collisions" column and with all 12 simulations listed. Same concerns all mentions of "six experiments" - there are 12 DNS runs.

We added Table 1 in the revision to summarize the model description.

To reduce the confusion, we defined the simulations without collisions as the "condensation-only" set and the ones with collisions as the "condensation-collision" set. And we updated the description of the two sets of experiments on lines 169-174:

"Two sets of experiments are performed. Each set consists of six cases, which gives 12 simulations in total. The first set of the experiment includes both condensational and collisional growth of droplets and will be referred to as the "condensation-collision" set. The second set excludes the droplet collision and will be referred to as the "condensation-only" set. The model setup for the two sets is the same other than the difference mentioned above. The configuration of the six cases is listed in Table 2. We focus on the condensation-collision set in the result section unless explicitly specified, and the condensation-only set is for the purpose of comparison to evaluate the influence by condensation and collision-coalescence. "

We modified the caption of Table 2 to:

"Model configuration of the six cases in each set of the experiment. Two sets of experiments are performed: set one includes both collision and condensation in the droplet growth and is referred to as the "condensation-collision" set; set two only considers droplet condensation and is referred to as the "condensation-only" set. This gives 12 cases in total. The natural DSD is taken from the parcel model output at S = 1.59%. Monodisperse seeding is considered in "seeded" cases with CCN size ( $R_d$ ) and initial droplet size (R) listed in the table."

8. Last but not least, please clarify if the study can be independently reproduced by providing information on the versions of the model code used and its availability.

We have added the GitHub links to the parcel model and DNS model used in this study in the *Code and data availability* section.

**Other remarks:**

• p1/l17: "interaction"  $\rightarrow$  interactions

**Checked**

• p1/l34: space before parenthesis missing

**Checked**

• p2/l23: framework  $\rightarrow$  frameworks

**Checked**

• p2/l26: "certain microphysical processes" – please be more specific

We removed the sentences and condensed the first paragraph in the introduction section to a more concise form.

• p2/l38: "solve"  $\rightarrow$  "solves"

**Checked**

• p2/l41: isn't the motivation to reduce the computational cost, rather than to reduce uncertainty? (replacing DNS with a parcel model actually increases uncertainty...), I would suggest removing the whole paragraph actually (lines 41–51)

The purpose of this parcel-DNS hybrid framework is to reduce the computational cost without sacrificing the accuracy of the physical representation of droplet growth. The hybrid method increases the accuracy in terms of modeling the impact of individual aerosols after the activation stage and reduces the computational cost by excluding the unactivated aerosols.

To clarify the motivation and reduce confusion, we have revised the paragraph as below (lines 39-45):

"Only a few DNS studies to date investigated the evolution of the droplet size distribution (DSD) in an updraft environment (e.g., Chen et al. 2018; Gotoh et al. 2016; Saito and Gotoh 2018). However, the solute effect (aerosol hygroscopicity) and curvature effect were excluded in those works for simplicity. Other DNS studies focused on the steady-state conditions, i.e., zero updrafts with zero mean supersaturation (e.g., Li et al. 2020; Sardina et al. 2015). It is recognized that DNS is computationally expensive. To achieve an accurate representation of cloud microphysics while maintaining a feasible computational load, a hybrid modeling framework that combines a parcel model and a DNS model is proposed in this study."

 p2/l45: "aerosol processing" in some contexts is used to refer to modification of ambient aerosol after evaporation of droplets (due to aqueous chemical reactions and collisions)
perhaps worth rephrasing

We have replaced the word "aerosol processing" with "aerosol activation".

• p3/l60: "nuclei ... enhances"  $\rightarrow$  "nuclei ... enhance" (or "representation of ...")

**Checked**

• p3/l69: "Section 2.1-2.2" → "Sections 2.1-2.2"

**Checked**

• p3/I73: "droplet chemistry composition"  $\rightarrow$  "hygroscopicity"

**Checked**

• p4/Fig2: suggest finding alternative wording for "stairs", please rephrase the last sentence: "fitting the distributions to the DNS" seems awkward, typo in "processers"

We replaced "stairs" with "histogram" for consistency across the entire article.

We removed the sentence from the figure and explained the treatment of assigning initial droplets in DNS in Section 2.2 (lines 140-147):

"... only the activated aerosols from the parcel model are carried over to the DNS, reducing the particle number concentration to  $N = 87 \ cm^{-3}$ . This treatment avoids the computation of tracking the inactivated particles. In the parcel model, the droplet size is calculated by using the moving-bin method. The dry radius of each bin remains constant, and the wet radius grows by condensation. To assign the initial droplet size and its dry radius in the DNS, we regrouped the activated droplet bins into 15 droplet size groups ( $R = 2 - 16 \ \mu m$ ) with an interval of  $1 \ \mu m$ . Their CCN sizes remain the original value. Due to the parallelization setup in the model, the initial number of each droplet size group has to be an exact multiple of the number of processors in the simulation (64 processors are used in the present simulations). Therefore, a small difference in the resulting DSD between the two models is expected, as shown in Fig. 2(c)."

• p5/l109: are four significant digits really necessary when specifying initial RH?

We agreed that the four digits are not necessary regarding the impact on the result. We put it this way to list our model configuration as it was.

• p5/l118: "thermodynamic equilibrium" sounds puzzling, I suggest following Jensen and Nugent and explaining what is meant: "in equilibrium (dr/dt=0)"

**Changed the statement**

• p6/l136: "aerosol processing" – see comment p2/l45 above

**Corrected**

 p6/eq2: drop growth equation (2) implies that supersaturation is defined as S = e/es (as in Jensen and Nugent 2007), but in Chen et al. 2018b it is defined as S = qv/qvs – of course numerically almost the same, but perhaps worth clarifying Added the definition when we introduced S below equation (1).

• p7/I167: why not replacing the inline fraction with just  $\kappa = 0$ ?

**Corrected**

• p7/l171: k→ к

**Corrected**

• p7/I166: "turbulent advection of the supersaturation fluctuation" suggests S' is among the advected quantities

It is correct that S is an advected quantity since S is determined by two passive scalars: T and Qv, both are advected by turbulence

• p8/184: "extremely slow": be more specific

We have modified the paragraph based on the new simulated results and thus removed this sentence.

• p8/187: "when"  $\rightarrow$  "When"

**Corrected**

• p8/l192: o() → *O*

**Corrected**

• p8/l197: space before parenthesis

**Checked**

• p8/l202: avoid word "claim"

**Changed it to "demonstrates"**

• p9/Fig2: mention in the caption that collisions were enabled

**Added**

- p11/Figs5-6 (Fig. 3 and Fig. 5 in the revision): mention in the caption that collisions were enabled
- p12/Fig7: mention in the caption that collisions were enabled

We explicitly mentioned in the captions that the three plots were based on results in the condensation-collision set of experiments.

• p14/l297-298: remove "which is a major facility"?

**Removed**

• p14/l300-301: rephrase "support from Cheyenne ... and from Graham and Cedar"

We rephrase the statement to "We would like to acknowledge high-performance computing (HPC) support from Cheyenne, Graham, and Cedar. HPC resources at Cheyenne (doi:10.5065/D6RX99HX) is provided by NCAR's Computational and Information Systems Laboratory and sponsored by the National Science Foundation. HPC resources at Graham and Cedar are provided by Compute Canada (www.computecanada.ca)."

• References: use journal abbreviations

**Checked**

- References: most entries have doi/url given twice
- References: if there is a doi assigned, do not list url (e.g.: Skamarock et al., Yang et al.)

**Removed the URL**

---

## Referee Report (RR1)

**Review of "Impact of hygroscopic CCN and turbulence on cloud droplet growth: A parcel-DNS approach" by Chen et al. (acp-2019-886)**

The revised study considers many of my previous concerns, and I feel much more comfortable with the results after the numerical issues have been resolved. However, there are still some issues that need to be addressed before publication. In particular, I feel that the aspect of "cloud seeding" requires some attention, as detailed below.

**Major Comments**

*Cloud seeding and ultra-giant nuclei*

First, I feel that the presumably overarching subject of "cloud seeding" is not well motivated. Cloud seeding is mentioned in the title and four times in the abstract, but it is only mentioned once at the end of the introduction (l. 79). Accordingly, the article lacks general background information on cloud seeding, e.g., how cloud seeding might be used to enhance precipitation in arid environments. Accordingly, I cannot find a continuous storyline that guides the reader through the manuscript, and the erratically occurring discussion of cloud seeding feels like a distraction. The authors owe their interesting results a more concise story. One leading idea for this study, which is discernable at some places already, might be the separation of the *artificial* influence of cloud seeding on the initiation of rain in comparison to *natural* cloud processes such as turbulence and the effect of aerosol hygroscopicity. In other words, one could simply ask: Does cloud seeding make sense?

Second, the initialization of giant aerosols ($R_d > 1$ µm) with their equilibrium radius (ll. 131 – 132) needs to be commented on. In subsaturated environments, these aerosols need several hours to days to reach their equilibrium radius (e.g., Mordy 1959), which indicates that these particles might have wet radii significantly smaller than their equilibrium radius when entering a cloud. Accordingly, these aerosols will not trigger the precipitation process as suspected by the authors (ll. 5 – 6). I was initially not too concerned with this shortcoming since the authors find that the contribution of these particles is insignificant (ll. 6 – 7). However, in the case of cloud seeding, the *natural* giant aerosols compete with the *artificially* added particles. For these cases, it might matter how large the largest natural particles are at the beginning of the simulation. And I suspect that smaller-than-equilibrium particles might increase the importance of seeded particles in the precipitation process, which will certainly affect statements such as ll. 282 – 283.

**Minor Comments**

Ll. 89 – 91: It feels redundant to mention twice that there is no more activation above the cloud base.

Ll. 89 – 91: It might be the case that there is no activation in the cloud core or your parcel model. However, activation of laterally entrained aerosols might occur in "more complete" cumulus clouds (Slawinska et al. 2012; Hoffmann et al. 2015).

Fig. 1: The supersaturation below cloud base increases approximately linear (see Fig. 2a). Please change this in your sketch.

Ll. 100 and 112 – 113: It feels redundant to mention twice that most droplets are smaller than 10 µm.

Ll. 166 – 167: Is there a significant effect of the fallen-out droplets on the LWC? Figure 7 a does not show a (significant) change in the increase of LWC once droplets reach 50 µm in radius. Similarly, the mean radius seems not to be affected by the fallen-out droplets. Could you please state the LWC of the fallen-out particles? This will give the reader a sense of how much water is lost due to precipitation.

Tab. 2, Figs. 3, 4, 5, 6, 7, and several places in the manuscript: When I previously suggested using more meaningful abbreviations for the individual model runs, I intended to use these abbreviations throughout the manuscript: Run NoTurb instead of Run B, Run NoSolu instead of Run C, …

Ll. 180 – 185: State explicitly that the seeded particles do not exhibit any variability in their initial radius, i.e., they are seeded using a delta distribution function.

L. 185: In addition to increasing the dry radius by a factor of 10, the wet radius in Run D3 is also doubled, according to Tab. 2.

L. 236: For clarity, add "non-turbulent" before "gravitational collection process" if appropriate.

Fig. 4: Panel "(b) Run C (cond+coll)" needs to be labeled "(c) Run C (cond+coll)".

Fig. 5: Change units on ordinate to "$cm^{-3}\,s^{-1}$"

L. 259: Consider moving the definition of the autoconversion rate from the caption of Fig. 7 to the main text.

L. 289: What is meant by "the inhibition effect"? I know that you are refereeing to the last sentence, but it is not obvious that the previous sentence describes the "inhibition effect".

Ll. 321 – 327: Since and the amount of water lost due to precipitation is presumably negligible (see also comment for ll. 166 – 167 above), and the range of tested aerosol concentrations and droplet sizes is rather small, it should not surprise that parcel-mean values "are not sensitive to turbulence level and aerosol conditions". It is expected that these quantities are approximately adiabatic. It might be interesting to add some lines to Fig. 7 showing the adiabatic LWC or the adiabatic mean radius, which will allow one to quantify the degree of non-adiabaticity in the conducted simulations.

**Technical Comments**
L. 13: "On the other hand" not "on the other hand"

L. 40 ff.: It is awkward to cite "Chen et al. 2018b" before "Chen et al. 2018a".

L. 85 ff.: Units should be displayed with upright characters, not italics.

L. 89: The SI symbol for seconds is "s" not "sec".

L. 111: There is a space missing between "flow." and "Studies".

L. 232 and other places: I suggest to abbreviate "minute" here because it is part of an equation (T = 6 min).

**References**
Hoffmann, F., Raasch, S., & Noh, Y. (2015). Entrainment of aerosols and their activation in a shallow cumulus cloud studied with a coupled LCM–LES approach. *Atmospheric Research*, *156*, 43-57.

Mordy, W. (1959). Computations of the growth by condensation of a population of cloud droplets. *Tellus*, *11*(1), 16-44.

Slawinska, J., Grabowski, W. W., Pawlowska, H., & Morrison, H. (2012). Droplet activation and mixing in large-eddy simulation of a shallow cumulus field. *Journal of the atmospheric sciences*, *69*(2), 444-462.

---

## Referee Report (RR2)

Comments to revised manuscript acp-2019-886
Jun 21 2020

**Reproducibility**

According to ACP Data Policy[1] "Data do not comprise the only information which is important in the context of reproducibility. Therefore, Copernicus Publications encourages authors to also deposit software, algorithms, model code, ... on suitable FAIR-aligned repositories/archives whenever possible."

   While the mentioned policy merely encourages to do so, let me urge the authors to do so, to do better than "parcel model and DNS model used to produce the dataset are available upon request" by making the code publicly available on a persistent repository (and/or as an electronic supplement to the paper).

**LWC profile and single/double precision issue**

The origin of the problem is somewhat surprising. Let me just note that in the reply to the first round of reviews, the authors made an apparent error in definition of the growth rate of particles expressed in the third power of radius. From the introduced equation for the growth rate $dR^2/dt$ it is evident that $Kf_vS$ has the unit of m$^2$/s, which is inconsistent with the later definition of $dR^3/dt$. It should read $dR^3/dt = 3RKf_v(S - f(solu, curv))$.

**Spectral discretisation**

The figure C provided in the reply to the first round of reviews confirms significant sensitivity to the spectral discretisation with low number of "moving bins". Consequently, it is hard to agree with the statement that "the resolution of the size spectrum will also have no numerical impact on the evolution of the droplet spectrum" (reply to the reviews). Please acknowledge the sensitivity in the text. I suggest including Fig. C in the manuscript.

**Lognormal spectrum initialisation**

The geometric standard deviation seems to be mismatched with the spread parameter (see caption of Table. 1 in Xue et al. 2010) as the geometric standard deviation must have values above unity. On a related note, doesn't the vertical axis of Fig. 2c denote $dN/d(ln(r))$ as in Fig. 4? Same concerns colour scale in Fig. 3.

**Supersaturation definition inconsistency**

Note that in order to arrive at eq. (1), one needs to define supersaturation as $S = e/e_s$ (passage from 7.15-7.16 to 7.17 in the Rogers & Yau book) – worth clarifying as the manuscript involves the alternative definition of $S$ using mixing ratios just below eq. (1).

**Seeding nomenclature**

Starting from the abstract, the manuscript mentions "seeding more aerosols", "seeding aerosol", "aerosols injected", etc; while in fact the seeded particles, given their size would classify as droplets being four or eight micron in radius. In the case of D1 and D2 runs, these particles are well above their critical sizes, hence would also classify as droplets because of being activated (the D3 seeding particles are below their critical sizes). To sum up, I suggest elaborating on the somewhat arbitrary choice of dry/wet sizes of the seeded particles, and switching from calling them "seeding aerosol" to "seeding particles".

   On a related note, "CCN-embedded droplets" and "aerosol-embedded cases" seem misleading, if not incorrect, to me. Please rephrase.
* * *
[1] https://www.atmospheric-chemistry-and-physics.net/about/data_policy.html

**Parcel-mean term**

For easier reading, I suggest refraining from using the "parcel-mean" term when referring to DNS simulations, to avoid confusion with the parcel model used for initialisation. Bulk or macroscopic might be better terms.

**Miscellaneous notes**

- p2/l28-29: the sentence "no benchmark "truth" from either measurements or modelling exists to gauge the performance of various microphysics schemes" arguably does not require the "Up to this date" opening – do the authors envisage that such "truth" will ever exist?;

- p2/l34-37: The mention of wall effects in cloud chambers in the very first paragraph of the introduction seem misplaced – suggest not to deviate too far from the scope of the paper in the introduction;

- p2/l74-75: suggest removing second part of the sentence ("to seamlessly ...");

- p3/l82: "ascending process" ⤳ "ascent";

- p4/l91: suggest removing "and aerosol activation is unimportant" (previous sentence mentions that no new activation occurs);

- p4/l92: "Outputs from the parcel model" ⤳ "Parcel model state"

- p6/l130-131: suggest removing the sentence "In this way, the numerical diffusion..." (out of scope);

- p6/l131: "at the given ..." – missing beginning of the sentence

- p6/Table 1: moving-bin method is contrasted with Lagrangian particle method – both are Lagrangian-in-particle-size; is there, in practice, any difference in the context of this work? Please elaborate or avoid contrasting them;

- p7/l156-161: background/motivation mixed with model description;

- p7/l164: "when small aerosols are introduced" – no small aerosols are introduced, right?

- p7/l172: correct "same other then";

- p9/l194: "aerosol ... broaden the DSD" – rephrase so that particular property of the aerosol population is mentioned

- p13/l179: ditto ("modulation by aerosols" – concentration, size, hygroscopicity?)

- p11/l251: please add a reference

- p14/Fig 7: x axis label missing in subplots (a) and (b)

- p16/l343: Lagrangian supersaturation: please elaborate

**Raster graphics**

Please replot all figures ensuring vector graphics format (not raster as it is currently provided)[2].

Hope that helps.
* * *
[2]see Manuscript composition: Figure composition in `https://www.atmospheric-chemistry-and-physics.net/for_authors/manuscript_preparation.html`

---

## Author Response (AR2)

We thank the two reviewers for their invaluable comments. We provided the point-by-point responses below. The comments are in black and the responses in blue. Changes made in the revised manuscript were also marked in blue.

**Comment # 1**

The revised study considers many of my previous concerns, and I feel much more comfortable with the results after the numerical issues have been resolved. However, there are still some issues that need to be addressed before publication. In particular, I feel that the aspect of "cloud seeding" requires some attention, as detailed below.

**Major Comments**

*Cloud seeding and ultra-giant nuclei*

First, I feel that the presumably overarching subject of "cloud seeding" is not well motivated. Cloud seeding is mentioned in the title and four times in the abstract, but it is only mentioned once at the end of the introduction (l. 79). Accordingly, the article lacks general background information on cloud seeding, e.g., how cloud seeding might be used to enhance precipitation in arid environments. Accordingly, I cannot find a continuous storyline that guides the reader through the manuscript, and the erratically occurring discussion of cloud seeding feels like a distraction. The authors owe their interesting results a more concise story. One leading idea for this study, which is discernable at some places already, might be the separation of the *artificial* influence of cloud seeding on the initiation of rain in comparison to *natural* cloud processes such as turbulence and the effect of aerosol hygroscopicity. In other words, one could simply ask: Does cloud seeding make sense?

We included motivation for using the idea of cloud seeding for studying the impact of aerosols on line 36-50:

"In this study, we implement the idea of in-cloud seeding, i.e., seeding hygroscopic particles near the cloud base to investigate the effects of aerosols in droplet growth and rain formation. Hygroscopic cloud seeding, owing to its potential effect of increasing rainfall, has been conducted in research and operational context globally to address the shortage of water resources in arid environments (e.g., Silverman and Sukarnjanaset, 2000; Terblanche et al., 2000). The general concept of hygroscopic cloud seeding in rain enhancement is that the introduction of artificial cloud condensation nuclei (seeding particles) into warm clouds can, on the one hand, suppress the activation of small natural aerosols, and on the other hand, generate large initial particles that accelerate or enhance the collision-coalescence process (Cooper et al., 1997). Regardless of its existence in operational weather modification for decades, the direct effect of seeding is still inconclusive, partly due to the chaotic nature of the convective cloud system making it impossible to conduct controllable seeding experiments and the limitation in detecting and assessing the seeding results with current instrumentations (Silverman, 2003; Flossmann et al., 2019). Nevertheless, the progress made in cloud seeding

does advance our understanding of cloud-aerosol-precipitation interactions. A leading idea of this study is to make use of the concept of cloud seeding experiments to separate the influence of aerosols on rain initiation  from the effects of natural cloud processes such as turbulence and aerosol hygroscopicity,  as well as to  shed light on the long-existing question of whether cloud seeding could enhance precipitation."

Second, the initialization of giant aerosols (Rd > 1 µm) with their equilibrium radius (ll. 131 – 132) needs to be commented on. In subsaturated environments, these aerosols need several hours to days to reach their equilibrium radius (e.g., Mordy 1959), which indicates that these particles might have wet radii significantly smaller than their equilibrium radius when entering a cloud. Accordingly, these aerosols will not trigger the precipitation process as suspected by the authors (ll. 5 – 6). I was initially not too concerned with this shortcoming since the authors find that the contribution of these particles is insignificant (ll. 6 – 7). However, in the case of cloud seeding, the *natural* giant aerosols compete with the *artificially* added particles. For these cases, it might matter how large the largest natural particles are at the beginning of the simulation. And I suspect that smaller-than-equilibrium particles might increase the importance of seeded particles in the precipitation process, which will certainly affect statements such as ll. 282 – 283.

Our original statement in the initialization of aerosol size was not correct. In our simulation, the parcel model had different treatments in computing the initial size below and above Rd = 1 µm. We only applied equilibrium radius to Rd < 1 µm, and for Rd >=1 µm we assumed a doubled volume in wet size, i.e., R = 2 ^ (⅓) Rd.
We have modified the original statement as following on line 154-156:
"For aerosols with dry radius Rd <= 1 µm, the initial wet radius is set to the size when the droplet is in equilibrium at the given ambient humidity: dR/dt=0 (Jensen and Nugent, 2017). For giant aerosols with Rd >1 µm, the initial wet size is assumed to be twice the dry volume, i.e., R = 2 ^ (⅓) Rd. "

It should be pointed out that the timescale of giant aerosols reaching an equilibrium size highly depends on the relative humidity. In a sub-saturated environment (initial relative humidity $RH_0$ = 85.61% in our case), the timescale is a few seconds. Fig. F below shows the parcel model results of the particle growth at $RH_0$ =85.61% (our case) and $RH_0$ =100%, respectively. In both cases, the dry radius is set to 10 µm with a number concentration of 1 $cm^{-3}$ , updraft velocity is set to 0 m/s (i.e., S is only modulated by droplet condensation). It is shown that in a sub-saturated environment, the time for the droplet to reach equilibrium is less than 20 seconds. In a saturated environment, it takes a few minutes to reach equilibrium. Overall, the higher the RH, the longer the aerosol takes to reach equilibrium size. But in both cases, they have a timescale shorter than an hour.

[Figure]

Fig. F. Radius varies with time with an initial $RH_0$ = 100.0% (red curve) and an initial $RH_0$ = 85.61% (blue curve).

**Minor Comments**

Ll. 89 – 91: It feels redundant to mention twice that there is no more activation above the cloud base.

Removed "and aerosol activation is unimportant"

Ll. 89 – 91: It might be the case that there is no activation in the cloud core or your parcel model. However, activation of laterally entrained aerosols might occur in "more complete" cumulus clouds (Slawinska et al. 2012; Hoffmann et al. 2015).

We have acknowledged this in the conclusion when addressing the shortcomings of the current study (line 357-360). We added the above-mentioned references in the revision.

Fig. 1: The supersaturation below cloud base increases approximately linear (see Fig. 2a). Please change this in your sketch.

Changed to linear increase

Ll. 100 and 112 – 113: It feels redundant to mention twice that most droplets are smaller than 10 µm.

Removed ", and most droplets in the parcel model are below 10 µm"

Ll. 166 – 167: Is there a significant effect of the fallen-out droplets on the LWC? Figure 7 a does not show a (significant) change in the increase of LWC once droplets reach 50 µm in radius. Similarly, the mean radius seems not to be affected by the fallen-out droplets. Could you please

state the LWC of the fallen-out particles? This will give the reader a sense of how much water is lost due to precipitation.

The fallout of R > 50 µm did not change the LWC in a discernible sense. Fig. G(a) shows the LWC including and excluding R > 50 µm. As the mass of R > 50 µm is about two orders of magnitude smaller than the LWC (Fig. G(b)), the two curves of LWCs are almost identical. Our study emphasized on the processes at the rain initiation stage. Therefore we mainly simulated the period of time before a substantial amount of precipitation forms.

[Figure]

*Fig. G: (a) The time series of LWC of the six simulations. Colors distinguish the cases. Solid lines denote the LWC including the mass of R > 50 µm and dashed lines represent the LWC excluding R>50 µm. (b) The time series of LWC of R > 50 µm.*

Tab. 2, Figs. 3, 4, 5, 6, 7, and several places in the manuscript: When I previously suggested using more meaningful abbreviations for the individual model runs, I intended to use these abbreviations throughout the manuscript: Run NoTurb instead of Run B, Run NoSolu instead of Run C, ...

Switched to meaningful abbreviations in the figs and throughout the manuscript.

Ll. 180 – 185: State explicitly that the seeded particles do not exhibit any variability in their initial radius, i.e., they are seeded using a delta distribution function.

We added "monodisperse" in the statement to stress the property of their size: "an extra number of **monodisperse** aerosols are introduced near the cloud base".

L. 185: In addition to increasing the dry radius by a factor of 10, the wet radius in Run D3 is also doubled, according to Tab. 2.

We have modified the sentence to "In Run D3, the dry size of the seeding particles increases to tenfold of that in Run D1 with the wet size doubled".

L. 236: For clarity, add "non-turbulent" before "gravitational collection process" if appropriate.

Modified to "non-turbulent gravitational collection process".

Fig. 4: Panel "(b) Run C (cond+coll)" needs to be labeled "(c) Run C (cond+coll)".

Corrected.

Fig. 5: Change units on ordinate to "cm^-3 s^-1"

Changed.

L. 259: Consider moving the definition of the autoconversion rate from the caption of Fig. 7 to the main text.

We have added the definition on line 276-277 while keeping the definition in the caption for clarification.

L. 289: What is meant by "the inhibition effect"? I know that you are referring to the last sentence, but it is not obvious that the previous sentence describes the "inhibition effect".

We have modified the sentence to "Increasing the size of seeding particles in Run D3 (Seed-1N2R) buffers the above-mentioned inhibition effect caused by increasing aerosol number concentration." And we also replaced "prohibited" by "inhibited" in the previous sentence.

Ll. 321 – 327: Since and the amount of water lost due to precipitation is presumably negligible (see also comment for ll. 166 – 167 above), and the range of tested aerosol concentrations and droplet sizes is rather small, it should not surprise that parcel-mean values "are not sensitive to turbulence level and aerosol conditions". It is expected that these quantities are approximately adiabatic. It might be interesting to add some lines to Fig. 7 showing the adiabatic LWC or the adiabatic mean radius, which will allow one to quantify the degree of non-adiabaticity in the conducted simulations.

Fig. F illustrated that the mean value is very close to the adiabatic value. Adding adiabatic LWC would complicate the figure. On line 255-256 we added: "This is because the fall out mass of drizzle drops of $R > 50 \ \mu m$ before T=500 s is negligible, and the bulk LWC of the six cases is approximately adiabatic." to explain the negligible effect of turbulence & aerosols on modulating the bulk LWC.

**Technical Comments**

L. 13: "On the other hand" not "on the other hand"

Corrected.

L. 40 ff.: It is awkward to cite "Chen et al. 2018b" before "Chen et al. 2018a".

The order of reference was automatically arranged by the ACP latex typesetting, which determines the index of "a" and "b" in the manuscript. I am not sure how to change that manually.

L. 85 ff.: Units should be displayed with upright characters, not italics.

It is also the default font in ACP latex format when using "$$" for numbers, units, and math symbols.

L. 89: The SI symbol for seconds is "s" not "sec".

Replaced all "sec" with "s" in the manuscript.

L. 111: There is a space missing between "flow." and "Studies".

Corrected.

L. 232 and other places: I suggest to abbreviate "minute" here because it is part of an equation (T = 6 min).

Has abbreviated "minute" to "min" in all equation forms.

**References**

Hoffmann, F., Raasch, S., & Noh, Y. (2015). Entrainment of aerosols and their activation in a shallow cumulus cloud studied with a coupled LCM–LES approach. *Atmospheric Research*, *156*, 43-57.

Mordy, W. (1959). Computations of the growth by condensation of a population of cloud droplets. *Tellus*, *11*(1), 16-44.

Slawinska, J., Grabowski, W. W., Pawlowska, H., & Morrison, H. (2012). Droplet activation and mixing in large-eddy simulation of a shallow cumulus field. *Journal of the atmospheric sciences*, *69*(2), 444- 462.

**Comment # 2**

**Reproducibility**

According to ACP Data Policy1 "Data do not comprise the only information which is important in the context of reproducibility. Therefore, Copernicus Publications encourages authors to also deposit software, algorithms, model code, ... on suitable FAIR-aligned repositories/archives whenever possible."

While the mentioned policy merely encourages to do so, let me urge the authors to do so, to do better than "parcel model and DNS model used to produce the dataset are available upon request" by making the code publicly available on a persistent repository (and/or as an electronic supplement to the paper).

We acknowledge that it is beneficial to share the model code with the readers if the code is in a user-friendly format. However, our model is a complex piece of code developed over two decades. Different parts of the code are written by different authors and a number of them are not co-authors of this paper. We simply do not have permission to share their part of the code with the general public. Additionally, the current version of the code is not well-documented, and it would be very difficult for general interested users to understand and to use it without a proper user's manual. Considerable effort is required to document the meaning of each variable, functionality of each module, etc. For these reasons, we do not feel proper to share the code with the general public at this moment.

**LWC profile and single/double precision issue**

The origin of the problem is somewhat surprising. Let me just note that in the reply to the first round of reviews, the authors made an apparent error in definition of the growth rate of particles expressed in the third power of radius. From the introduced equation for the growth rate dR2/dt it is evident that KfvS has the unit of m2/s, which is inconsistent with the later definition of dR3/dt. It should read dR3/dt = 3RKfv(S − f(solu, curv)).

We apologize for the typo in the first round of review, and the model code used the right format of the equation. The correct format of the equation in $R^3$ form should be $\frac{dR^3}{dt} = 3RKf_v(S - f(solu, curv))$, as addressed by the reviewer.

**Spectral discretisation**

The figure C provided in the reply to the first round of reviews confirms significant sensitivity to the spectral discretisation with low number of "moving bins". Consequently, it is hard to agree with the statement that "the resolution of the size spectrum will also have no numerical impact

on the evolution of the droplet spectrum" (reply to the reviews). Please acknowledge the sensitivity in the text. I suggest including Fig. C in the manuscript.

Based on the sensitivity test, the relative error caused by changing bin resolution is below 2.3% of the total aerosol number concentration ( $2.6 cm^{-3}$ higher than the 253-bin case). In particular, the 39-bin case is only 0.6 $cm^{-3}$ lower than the 253-bin case. Therefore, we think the sensitivity is not significant.

[Figure]

Fig. D. *Relative variation* $= (N_{act} - N_{act0})/N_{tot}$ , *where* $N_{act}$ *is the number of activated aerosols at each bin resolution,* $N_{act0}$ *is the number of activated aerosols at 253-bin, and* $N_{tot}$ *is the total number of aerosols.*

We have included the sensitivity analysis in the text.
"The result shows that the variation caused by changing bin resolution has a decreasing trend with increasing resolution, with a maximum variation of 2.3% of the total aerosol number concentration in the 32-bin case. In particular, the 39-bin case has only $0.6\ cm^{-3}$ more aerosols activated than in the 253-bin case."

**Lognormal spectrum initialisation**

The geometric standard deviation seems to be mismatched with the spread parameter (see caption of Table. 1 in Xue et al. 2010) as the geometric standard deviation must have values above unity. On a related note, doesn't the vertical axis of Fig. 2c denote dN/d(ln(r)) as in Fig. 4? Same concerns colour scale in Fig. 3.

The geometric standard deviation $\sigma$ of the maritime particle size distribution in Xue et al. 2010 was originally taken from Jaenicke (1988) with $\sigma$ value taken as the logarithm with base 10: $\sigma' = log10(\sigma) = [0.657\ 0.210\ 0.396]$ , and it was then converted to the natural logarithm value (see

footnote b of Table. 1): $\sigma' = log10(\sigma) = [1.5128\ 0.4835\ 0.9118]$. To avoid confusion, we changed to its original value $\sigma = [4.5394\ 1.6218\ 2.4889]$.

The vertical axis in Fig. 2c denotes the number concentration of each assigned bin in the model, for easier comparison between the parcel model and DNS. Therefore, it will be N. We added "The vertical axis denotes the number concentration of the assigned particle size in the model." in the caption of Fig. 2 for clarification.

We corrected the unit in Fig. 3 to "$dN/d(ln(r))\ (cm^{-3}\mu m^{-1})$"

**Supersaturation definition inconsistency**

Note that in order to arrive at eq. (1), one needs to define supersaturation as S = e/es (passage from 7.15-7.16 to 7.17 in the Rogers & Yau book) – worth clarifying as the manuscript involves the alternative definition of S using mixing ratios just below eq. (1).

On the one hand, it is more convenient to use mixing ratio ($q_v$) instead of vapor pressure (e) for calculating supersaturation, because qv is a predicted variable of the model. On the other hand, S of the atmosphere using mixing-ratio is a very good approximation to the original form. $q_v$ is defined as $q_v = \varepsilon\frac{e}{P-e}$ (eq. (2.18) in Rogers & Yau 1989), where $\varepsilon \approx 0.615$. One can derive $\frac{q_v}{q_{vs}} = \frac{e/(P-e)}{e_s/(P-e_s)} \approx \frac{e}{e_s}$, given $e \ll P$ and $e_s \ll P$ in the atmosphere. In our simulation, the relative error is below 0.02% (Fig. E below), which is negligible.

[Figure]

Fig. E. The relative error in calculating supersaturation due to using mixing ratio.

**Seeding nomenclature**

Starting from the abstract, the manuscript mentions "seeding more aerosols", "seeding aerosol", "aerosols injected", etc; while in fact the seeded particles, given their size would classify as

droplets being four or eight micron in radius. In the case of D1 and D2 runs, these particles are well above their critical sizes, hence would also classify as droplets because of being activated (the D3 seeding particles are below their critical sizes). To sum up, I suggest elaborating on the somewhat arbitrary choice of dry/wet sizes of the seeded particles, and switching from calling them "seeding aerosol" to "seeding particles".

Used the nomenclature "seeding particles" throughout the manuscript.

On a related note, "CCN-embedded droplets" and "aerosol-embedded cases" seem misleading, if not incorrect, to me. Please rephrase.

We rephrased it to "solute-containing droplets" instead of "CCN-embedded" or "aerosol-embedded".

Parcel-mean term

For easier reading, I suggest refraining from using the "parcel-mean" term when referring to DNS simulations, to avoid confusion with the parcel model used for initialisation. Bulk or macroscopic might be better terms.

Replaced "parcel-mean" with "bulk".

Miscellaneous notes

p2/l28-29: the sentence "no benchmark "truth" from either measurements or modelling exists to gauge the performance of various microphysics schemes" arguably does not require the "Up to this date" opening – do the authors envisage that such "truth" will ever exist?;

Removed from the sentence

p2/l34-37: The mention of wall effects in cloud chambers in the very first paragraph of the introduction seem misplaced – suggest not to deviate too far from the scope of the paper in the introduction;

We have removed the sentence of "For example, Thomas et al. (2019) used a flux-balance model to estimate the wall effect on the mean temperature and mean water vapor mixing ratio and found that the results highly depend on the geometry of the chamber."

p2/l74-75: suggest removing second part of the sentence ("to seamlessly ...");

Removed
p3/l82: "ascending process" Y "ascent";

Replaced with "ascent"

p4/l91: suggest removing "and aerosol activation is unimportant" (previous sentence mentions that no new activation occurs);

Removed

p4/l92: "Outputs from the parcel model" Y "Parcel model state"

Changed as suggested.

p6/l130-131: suggest removing the sentence "In this way, the numerical diffusion..." (out of scope);

Removed

p6/l131: "at the given ..." – missing beginning of the sentence

Changed to "The initial wet size is set to the size when droplet is in equilibrium at the given ambient humidity: $dR/dt = 0$ (Jensen and Nugent, 2017)."

p6/Table 1: moving-bin method is contrasted with Lagrangian particle method – both are Lagrangian- in-particle-size; is there, in practice, any difference in the context of this work? Please elaborate or avoid contrasting them;

We changed the method name to "Lagrangian particle-by-particle method" to distinguish the two methods.

p7/l156-161: background/motivation mixed with model description;

We moved the motivation to line 55-61 in the introduction section.

p7/l164: "when small aerosols are introduced" – no small aerosols are introduced, right?

We changed to "when small droplets are present" for clarification.

p7/l172: correct "same other then";

We believe "other than" is the right spelling.

p9/l194: "aerosol ... broaden the DSD" – rephrase so that particular property of the aerosol population is mentioned

We changed it to "including solute and turbulence effectively broaden the DSD at different times" to reflect the comparison between the control case (Run A), no turbulence case (Run B), and no solute case (Run C).

p13/l279: ditto ("modulation by aerosols" – concentration, size, hygroscopicity?)

We changed it to "the modulation of droplet mean radius by seeding particles is larger than the modulation by collision-coalescence" to reflect that the modulation was made by introducing additional aerosols of different sizes and of different number concentration (seeding particles).

p11/l251: please add a reference

We have removed the paragraph as the information is inaccurate. The theoretical study by Liu et al. (2006, Fig. 1) and observations by Hudson and Yum (1997) demonstrated that for particle number concentration around $100 \, cm^{-3}$ and mean radius between 5-15 $\mu m$, the relative dispersion ranges from 0.01-0.1. The value of our cases at the end of the simulation is also from 0.01-0.1, consistent with those studies. We, therefore, added "The values among the six cases at the end of the simulation range from 0.01-0.1, which is highly consistent with the theoretical study by Liu et al. (2006, Fig. 1) for an aerosol number concentration close to $100 \, cm^{-3}$." on lines 261-262.

p14/Fig 7: x axis label missing in subplots (a) and (b)

x labels added.

p16/l343: Lagrangian supersaturation: please elaborate

Modified to "a highly perturbed Lagrangian **history of** supersaturation experienced by droplets."

Raster graphics

Please replot all figures ensuring vector graphics format (not raster as it is currently provided). Hope that helps.

Changed to vector graphics